# Scalable and General Whole-Body Control for Cross-Humanoid Locomotion

**Yufei Xue** [*] [1] [2]   **Yunfeng Lin** [*] [1] [2]   **Wentao Dong** [1] [2]   **Yang Tang** [1] [2]   **Jingbo Wang** [2]   **Jiangmiao Pang** [2]   **Ming Zhou** [2]
**Minghuan Liu** [1]   **Weinan Zhang** [1] [2]

https://xhugwbc.github.io

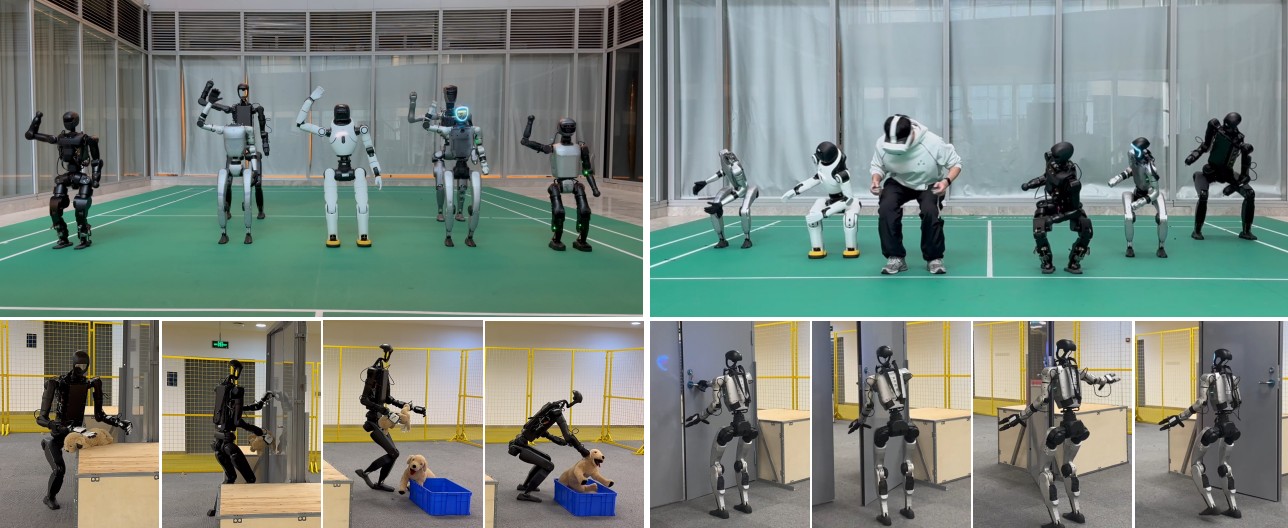

*Figure 1.* **Zero-shot generalization and real-world humanoid capabilities enabled by XHugWBC's generalist policy. First row:** Robust zero-shot generalization across seven humanoids with diverse DoFs, dynamic characteristics, and morphological structures. **Second row:** flexible teleoperation using XHugWBC enables long-horizon whole-body loco-manipulation tasks.

## Abstract

Learning-based whole-body controllers have become a key driver for humanoid robots, yet most existing approaches require robot-specific training. In this paper, we study the problem of cross-embodiment humanoid control and show that a single policy can robustly generalize across a wide range of humanoid robot designs with one-time training. We introduce XHugWBC, a novel cross-embodiment training framework that enables generalist humanoid control through: (1) physics-consistent morphological randomization, (2) semantically aligned observation and action spaces across diverse humanoid robots, and (3) effective policy architectures modeling morphological

and dynamical properties. XHugWBC is not tied to any specific robot. Instead, it internalizes a broad distribution of morphological and dynamical characteristics during training. By learning motion priors from diverse randomized embodiments, the policy acquires a strong structural bias that supports zero-shot transfer to previously unseen robots. Experiments on twelve simulated humanoids and seven real-world robots demonstrate the strong generalization and robustness of the resulting universal controller.

## 1. Introduction

Learning-based whole-body control (WBC) has become a dominant paradigm for legged robots, especially humanoids, enabling agile and robust behaviors beyond conventional model-based controllers (Cheng et al., 2024a; He et al., 2024; Liao et al., 2025; Xue et al., 2025; Yang et al., 2025a; Li et al., 2025). However, most existing controllers target a single embodiment (Pan et al., 2025; Chen et al., 2025b), requiring extensive retraining when transferred to new platforms. This limits scalability as each new robot differs in

* Equal contributions: Yufei Xue and Yunfeng Lin. [1]Shanghai Jiao Tong University [2]Shanghai AI Laboratory. Correspondence to: Weinan Zhang <wnzhang@sjtu.edu.cn>, Minghuan Liu <minghuanliu@sjtu.edu.cn>, Ming Zhou <zhouming@pjlab.org.cn>.

*Proceedings of the $43^{rd}$ International Conference on Machine Learning, Seoul, South Korea. PMLR 306, 2026. Copyright 2026 by the author(s).*

morphology, kinematics, and dynamics, making per-robot training costly and inefficient. A key open question therefore remains: **can a single learned controller generalize across diverse humanoid embodiments?**

Achieving this is challenging due to highly dynamic contacts, strict stability requirements, and large morphological variations. Prior cross-embodiment efforts in animation (Gupta et al., 2022; Huang et al., 2020) and locomotion (Yang et al., 2025b; Liu et al., 2025a) rely on shared morphological priors, unified state-action representations, or similar dynamics. Such assumptions break down for humanoids, which diverge substantially in kinematic structure, degrees of freedom, joint ordering, and physical properties. Simplifications that have facilitated transfer in quadrupeds such as reduced dynamic models or relatively homogeneous body morphologies also break down for humanoids (Feng et al., 2023; Luo et al., 2024), whose morphological diversity and whole-body dynamics are intrinsically more complex.

To address these challenges, we propose `XHugWBC`, a framework for learning a cross-humanoid whole-body controller. `XHugWBC` combines: 1) physically consistent morphological randomization, 2) a unified state-action representation that semantically aligns different embodiments, and 3) a policy architecture modeling embodiment-specific and graph-based representations derived from robot topologies.

We evaluate `XHugWBC` across diverse humanoids in simulation and reality. A single generalist policy trained with our framework achieves robust zero-shot whole-body control across seven real-world humanoid robots. In simulation, it scales to twelve distinct embodiments, reaching about 85% of specialist performance, while generalist-initialized fine-tuning surpasses specialists by up to 10%. These results show that `XHugWBC` learns strong embodiment-agnostic motion priors and enables scalable, general-purpose humanoid control. Our main contributions are:

- A physics-consistent morphological randomization that yields diverse and physically meaningful embodiments.

- A universal embodiment representation with tailored training techniques for cross-humanoid whole-body control.

- To the best of our knowledge, this is the first generalist controller demonstrating robust zero-shot whole-body control across seven real-world humanoid robots with substantial diversity.

## 2. Related Work

### 2.1. Cross-Embodiment Learning for Legged Robots

Cross-embodiment learning aims to find control policies that generalize across robots with different morphologies and dynamics, reducing the need for embodiment-specific controllers. Most existing approaches train on a small set of embodiments and achieve limited transfer (Yang et al., 2025b; Lin et al., 2025; Peng et al., 2026; Doshi et al., 2024), or introduce diversity through simple morphological randomization (Yu et al., 2023; Luo et al., 2024; Liu et al., 2025a; Ai et al., 2025). These methods typically assume aligned state-action spaces and similar embodiment dynamics, restricting their applicability to narrower robot families.

Humanoid robots, however, exhibit heterogeneous structures, physical properties and state representations, thus violating such assumptions. To address this, some efforts resort to biology-inspired trajectory generators (Shafiee et al., 2024), while others represent embodiments with joint-level descriptor sequences (Bohlinger et al., 2024; Ai et al., 2025). In contrast, `XHugWBC` generates physically consistent embodiment data and leverages a hybrid-mask Transformer to capture broad humanoid motion priors, enabling robust generalization with a single policy.

### 2.2. Humanoid Whole-Body Control

Whole-body control is a critical yet challenging task for humanoid robot learning and real-world applications. Recent work has explored unified command spaces (Xue et al., 2025; Li et al., 2025; Sun et al., 2025; Zhang et al., 2026), typically using command sampling and reinforcement learning. Other studies focus on motion tracking and representation learning from individual demonstrations (Xie et al., 2025; He et al., 2025; Huang et al., 2025) or large-scale datasets (Yin et al., 2025; Zhang et al., 2025; Ze et al., 2025; Luo et al., 2025), but often assume fixed robot morphologies. Additional progress has been made in whole-body loco-manipulation within controlled environments (Wang et al., 2025b; Weng et al., 2025; Su et al., 2025; Chen et al., 2025a; Zhuang & Zhao, 2025).

Our work follows Xue et al. (2025) and adopts a unified command space to enable expressive whole-body behaviors while explicitly targeting cross-humanoid generalization.

## 3. Method

### 3.1. Physics-Consistent Morphological Randomization

With the success of domain randomization (Peng et al., 2018; Hwangbo et al., 2019; Miki et al., 2022; Wu et al., 2023) in training robust whole-body controllers (Xue et al., 2025; Zeng et al., 2025; Wang et al., 2025a), a straightforward approach for learning cross-humanoid policies is to randomize embodied data (Feng et al., 2023; Bohlinger et al., 2024) across kinematic, dynamic, and morphological dimensions. However, a robot's inertial parameters must satisfy physical consistency (Traversaro et al., 2016) to represent plausible rigid bodies, making the feasible parame-

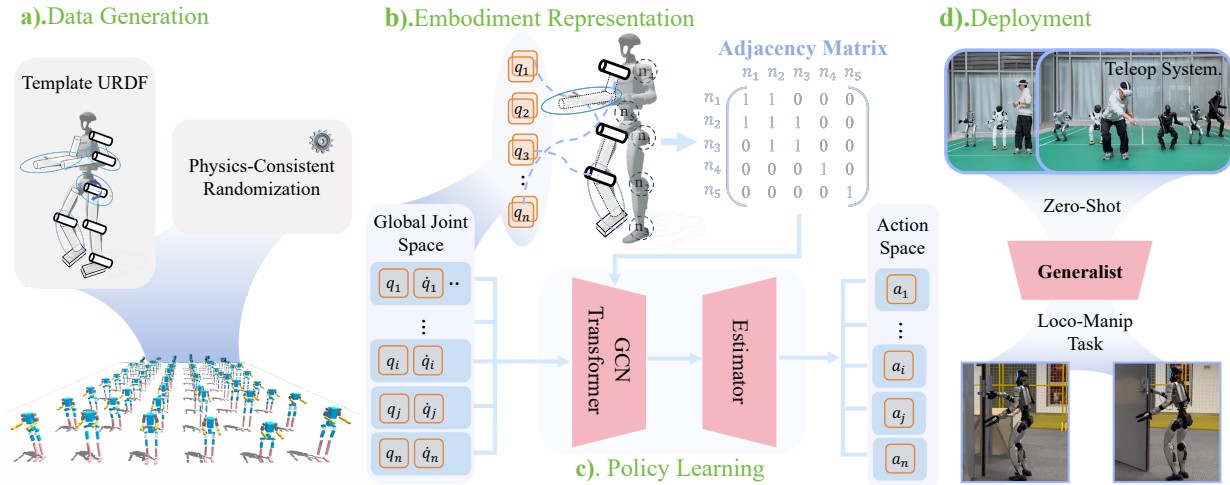

*Figure 2.* **Training framework of XHugWBC.** (a) **Data generation**: physics-consistent morphological randomization produces diverse and physically meaningful embodiments. (b) **Universal embodiment representation**: robot-specific states are projected into a global joint space, upon which an embodiment graph is constructed. (c) **Policy learning**: the generalist policy uses a GCN- or Transformer-based encoder together with a state estimator. (d) **Deployment**: the learned policy generalizes to seven humanoid robots with different kinematic, dynamic, and morphological structures in zero-shot.

ter space inherently non-convex and discontinuous. Prior work on physics-based simulation and legged locomotion often overlooks this requirement, either because experiments are conducted entirely in simulation or because robot links are approximated using simple geometric primitives whose dynamics are scaled in a linear and often physically invalid manner (Trabucco et al., 2022; Ai et al., 2025). For humanoid robots with highly complex and heterogeneous structures, perturbing inertial parameters without enforcing physical consistency can easily produce unrealizable models, destabilize simulation, drive policy optimization toward degeneracy, and ultimately cause catastrophic failures in sim-to-real transfer. We therefore introduce physics-consistent morphological randomization to generate diverse and plausible embodied data.

**Parameterizing a template robot.** We capture the shared structures of humanoid robots in a template model and define a parameter vector that encodes its kinematic layout, link geometries, inertial properties, and joint configurations.

$$\boldsymbol{\kappa} = [\boldsymbol{\kappa}_{\text{link}}, \boldsymbol{\kappa}_{\text{joint}}] \in \mathbb{R}^N \ ,$$

where $\boldsymbol{\kappa}_{\text{link}} \in \mathbb{R}^{10n_b}$ are parameters of robot links, $\boldsymbol{\kappa}_{\text{joint}} \in \mathbb{R}^{13n_d}$ are parameters of robot joints, $n_b \in \mathbb{N}_+$ and $n_d \in \mathbb{N}_+$ are the number of the rigid bodies and the number of degrees of freedom, respectively. In detail, $\boldsymbol{\kappa}_{\text{link}} = [\boldsymbol{\kappa}_{\text{l}}^{1\top}, \ldots, \boldsymbol{\kappa}_{\text{l}}^{n_b\top}]^\top$ represents the contributions from each rigid body, each $\boldsymbol{\kappa}_{\text{l}}^{i}$ corresponds to the inertial parameters of the $i$-th rigid body and is composed of

$$[m, h_x, h_y, h_z, I_{xx}, I_{yy}, I_{zz}, I_{xy}, I_{xz}, I_{yz}]^\top \in \mathbb{R}^{10}$$

where $m$ denotes the mass, $\mathbf{h} = [h_x, h_y, h_z]^\top = m\mathbf{c}$ is the first mass moments, where $\mathbf{c} \in \mathbb{R}^3$ denotes the center-of-

mass (CoM) coordinates in the body frame, and

$$\bar{\mathbf{I}} = \begin{bmatrix} I_{xx} & I_{xy} & I_{xz} \\ I_{xy} & I_{yy} & I_{yz} \\ I_{xz} & I_{yz} & I_{zz} \end{bmatrix}$$

is the rotational inertia w.r.t. the coordinate origin. Similarly, joint parameters $\boldsymbol{\kappa}_{\text{joint}}$ represent the contributions from each joint and can be expressed as $\boldsymbol{\kappa}_{\text{joint}} = [\boldsymbol{\kappa}_{\text{j}}^{1\top}, \ldots, \boldsymbol{\kappa}_{\text{j}}^{n_d\top}]^\top$, where each $\boldsymbol{\kappa}_{\text{j}}^{i}$ connects a parent link and a child link, and is defined by both spatial and dynamic parameters

$$[p_x, p_y, p_z, \phi, \theta, \psi, a_x, a_y, a_z, q_{\min}, q_{\max}, \dot{q}_{\max}, \tau_{\max}]^\top \in \mathbb{R}^{13}$$

where the $\mathbf{p} = [p_x, p_y, p_z]$ specifies the joint position in the parent frame, $\mathbf{e} = [\phi, \theta, \psi]$ represent its orientation, $\mathbf{a} = [a_x, a_y, a_z]$ defines the motion axis, and $q_{\min}$, $q_{\max}$, $\dot{q}_{\max}$ and $\tau_{\max}$ denote the range of motion, maximum velocity and torque, respectively. For randomizing morphologies, intuitively, we can apply perturbations within this morphological space of the template robot:

$$\boldsymbol{\kappa}' = \boldsymbol{\kappa} + \Delta\boldsymbol{\kappa} \ , \quad \Delta\boldsymbol{\kappa} \sim \mathcal{D}_{\text{morph}} \ . \tag{1}$$

where $\mathcal{D}_{\text{morph}}$ is noise distribution. Nevertheless, adding $\Delta\boldsymbol{\kappa}$ is non-trivial, as arbitrary noise can break physical consistency. To achieve reasonable morphology randomization, we reparameterize the morphological space into a form that we can easily add noise to.

**Reparameterizing the link space.** We first derive a physics-consistent randomization solution for the link space, which mainly consists of inertial parameters.

**Definition 3.1** (Physics-Consistent Inertial Parameters). A rigid body's inertial parameters $\kappa_{\text{link}}$ are said to be physically consistent if its pseudo-inertia matrix

$$\mathbf{J} = \begin{bmatrix} \mathbf{\Sigma} & \mathbf{h}^\top \\ \mathbf{h} & m \end{bmatrix} \tag{2}$$

is symmetric positive definite, where $\mathbf{\Sigma} = \frac{1}{2}\text{Tr}(\bar{\mathbf{I}})\mathbf{I} - \bar{\mathbf{I}}$.

Recall that the pseudo-inertia matrix admits the integral form (Traversaro et al., 2016) as

$$\mathbf{J} = \int_V \mathbf{q}\mathbf{q}^\top \rho(\mathbf{x})\,dV, \quad \mathbf{q} = [\mathbf{x}^\top, 1]^\top, \tag{3}$$

and the positive definiteness constraint $\mathbf{J} \succ 0$ can be enforced via linear matrix inequalities (Rucker et al., 2022).

**Lemma 3.2** (Cholesky-Level Parameterization). *If* $\mathbf{J} \succ 0$, *then there exists a unique upper-triangular matrix* $\mathbf{L}$ *with positive diagonal entries such that*

$$\mathbf{J} = \mathbf{L}\mathbf{L}^\top. \tag{4}$$

Lemma 3.2 implies that any physically consistent inertia matrix can be smoothly parameterized by its Cholesky factor $\mathbf{L}$. Therefore, physics-consistent randomization can be implemented by perturbing $\mathbf{L}$ as

$$\mathbf{L}' = \mathbf{L} + \boldsymbol{\epsilon}, \quad \boldsymbol{\epsilon} \sim \mathcal{D}, \quad \mathbf{J}' = \mathbf{L}'\mathbf{L}'^\top, \tag{5}$$

where $\mathcal{D}$ denotes a noise distribution. To provide geometric interpretability of such perturbations, we further examine how the pseudo-inertia transforms under affine deformations of the rigid body.

**Lemma 3.3** (Affine Transformation of Pseudo-Inertia). *Under an affine transformation of homogeneous body coordinates* $\mathbf{q}' = \mathbf{E}\mathbf{q}$ *and a mass density scaling* $\rho' = \beta^2\rho$, *the pseudo-inertia matrix transforms as*

$$\mathbf{J}' = \int_V \mathbf{E}\mathbf{q}\mathbf{q}^\top \mathbf{E}^\top \beta^2 \rho(\mathbf{x})\,dV = \mathbf{U}\mathbf{J}\mathbf{U}^\top, \tag{6}$$

*where* $\mathbf{U} = \beta\mathbf{E}$ *and* $\mathbf{q} = [\mathbf{x}, 1]^\top$.

Combining Lemma 3.2 and Lemma 3.3, any physically consistent inertia perturbation can be expressed as

$$\mathbf{J}' = (\mathbf{U}\mathbf{L})(\mathbf{U}\mathbf{L})^\top. \tag{7}$$

**Lemma 3.4** ($\mathbb{R}^{10}$ Bijective Mapping of Inertia). *The transformation matrix* $\mathbf{U} \in GL^+(4)$ *admits a unique upper-triangular factorization with positive diagonal entries. Moreover, parameterizing the diagonal entries via exponential maps yields a bijection between such* $\mathbf{U}$ *and a 10-dimensional real vector*

$$\theta_{inert} = [\alpha, d_1, d_2, d_3, s_{12}, s_{23}, s_{13}, t_1, t_2, t_3]^\top \in \mathbb{R}^{10}, \tag{8}$$

*where the explicit matrix form of* $\mathbf{U}$ *is given in Appendix A.1.*

**Proposition 3.5** (Smooth Physics-Consistent Randomization). *Under Lemma 3.2–3.4, any physically consistent inertia perturbation* $\mathbf{J}' \succ 0$ *can be written as*

$$\mathbf{J}' = (\mathbf{U}\mathbf{L})(\mathbf{U}\mathbf{L})^\top, \tag{9}$$

*where* $\mathbf{U}$ *is uniquely determined by* $\theta_{inert} \in \mathbb{R}^{10}$.

Proof of proposition 3.5 is in Appendix A.1. Consequently, physics-consistent randomization of inertial parameters can be performed by unconstrained perturbations in $\mathbb{R}^{10}$, while preserving physical feasibility and smoothness of the parameter space.

**Reparameterizing the joint space.** joint parameters are typically constrained by the overall physical and structural design of the robot. Therefore, we perform joint space randomization over *joint rotational axes*, *joint position*, and *joint actuation*. In particular, 1) we randomize the orientation of the hip joint's rotational axis $\mathbf{a} = [a_x, a_y, a_z]$, which enables the synthesis of a wide range of kinematically distinct hip configurations. The remaining joints follow a similar arrangement sequence across platforms. 2) We randomize joint position $\mathbf{p} = [p_x, p_y, p_z]$ relative to the parent link's frame, with magnitudes bounded within twice the distance to the link's center of mass. For motor control, the proportional-derivative (PD) gains and torque limits $\tau_{\max}$ are linearly scaled with the total mass of the robot as a heuristic approximation to maintain consistent actuation behavior across morphologies. 3) We randomize the actuation type by sampling selected joints as either *revolute* or *fixed*, where fixed joints are rigidly locked and excluded from control. Joints subject to this randomization include three waist joints (roll, yaw, and pitch), seven arm joints (shoulder roll, yaw, and pitch; elbow pitch; and wrist roll, yaw, and pitch), and three head joints (roll, yaw, and pitch). As a result, the controller can operate on robots with widely varying degrees of freedom, ranging from a minimum of 12 active joints, corresponding to a pure bipedal configuration with three hip joints (roll, yaw, pitch), one knee joint, and two ankle joints (pitch, roll) per leg, to a maximum of 32 active joints, which additionally include three waist joints, two arms with seven joints each, and three head joints. Detailed randomization ranges are provided in Appendix A.2.

### 3.2. Universal Cross-Embodiment Representation

Our goal is to develop a unified control policy across various morphologically diverse humanoid robots, each characterized by unique kinematic morphologies and differing numbers of actuated joints. A key challenge lies in the significant heterogeneity in their respective state and action spaces. To address this, we map robot-specific joint states into a global joint space for semantic alignment, and construct an embodiment graph on top of this global space to explicitly encode morphological structure.

**Joint space semantic alignment.** (Lin et al., 2025) proposes a hardware-agnostic joint space mapping that standardizes joint representations across embodiments, enabling a consistent robot control interface. Following this idea, we define a canonical joint dimension of $N_{\max} = 32$, where each index corresponds to a semantically aligned joint in a globally defined ordering. For any humanoid robot, its joints are embedded into this canonical space according to their kinematic roles and semantic identities. Formally, for a robot instance with $N_r \leq N_{\max}$ joints and their configuration $\mathbf{q}_r \in \mathbb{R}^{N_r}$, we define a mapping

$$\phi_r : \mathbb{R}^{N_r} \to \mathbb{R}^{N_{max}},$$

such that the canonical joint state $\mathbf{q}_{\text{global}}$ is constructed as:

$$\mathbf{q}_{\text{global}}[i] = \begin{cases} \mathbf{q}_r^{(j)}, & \text{if physical joint } j \text{ maps to global joint } i, \\ 0, & \text{otherwise.} \end{cases}$$
(10)

The resulting $\mathbf{q}_{\text{global}} \in \mathbb{R}^{N_{\max}}$ is a zero-padded, canonical representation shared across all embodiments. By enforcing semantic alignment at the joint level, the control policy receives fixed-dimensional inputs regardless of morphological variations or differences in actuation. The global joint index definition is provided in Table 8.

**Graph-based morphology description.** The unified representation above naturally supports constructing a graph-based description of each robot's morphology. Each embodiment represented by $\mathbf{q}_{\text{global}}$ can be converted to a directed kinematic graph: $\mathcal{G} = (\mathcal{V}, \mathcal{E})$, where the vertex set $\mathcal{V} = (v_1, v_2, \ldots, v_i)$ corresponds to joints, and the edge set

$$\mathcal{E} = \{e_{ij} = (v_i, v_j) \mid v_i, v_j \in \mathcal{V}\} \subseteq \mathcal{V} \times \mathcal{V}$$

captures rigid-body connections, where each edge $e_{ij}$ encodes the linkage between joint $v_i$ and $v_j$. The adjacency matrix $\mathbf{A} \in \{0,1\}^{N_{\max} \times N_{\max}}$ is constructed from $\mathcal{E}$ and encodes the overall connectivity of the embodiment:

$$(\mathbf{A})_{ij} = \begin{cases} 1, & \text{if } (v_i, v_j) \in \mathcal{E}, \\ 0, & \text{otherwise.} \end{cases}$$
(11)

Humanoid robots often adopt parallel-linkage mechanisms, which complicate graph construction. To address this, we collapse all nodes involved in a parallel linkage and connect them directly to the preceding joint in the articulation. For instance, ankle joints are treated as immediate children of the knee joint in the resulting graph. The final graph $\mathcal{G}$ for each robot is connected and acyclic, forming a kinematic tree whose vertices may represent either actuated or fixed joints. This structure is expressive enough to encode diverse kinematic and actuation patterns, as well as morphological information for general control. Figure 2 (b) illustrates the mapping from the joint-level description to its corresponding adjacency matrix.

## 3.3. Cross-Humanoid Learning

With physics-consistent morphological randomization and universal representation of structurally distinct robots, we proceed to develop a training strategy for a cross-humanoid control policy. We formulate the problem as a reinforcement learning task defined over a family of morphologically diverse robots $k \sim \mathcal{K}$. Each robot instance $k$ is modeled as a Partially Observable Markov Decision Process (POMDP):

$$\mathcal{P}_k = (\mathcal{S}_k, \mathcal{O}_k, \mathcal{A}_k, \mathcal{R}_k, \gamma),$$

where $\mathcal{S}_k$, $\mathcal{O}_k$ and $\mathcal{A}_k$ denote the state, observation, and action subspaces of robot $k$, represented in a universal space. $\mathcal{R}_k$ is the reward function, and $\gamma$ is the discount factor. The learning objective is to optimize a single policy that maximizes the expected return over all embodiments:

$$\max_{\pi} \mathbb{E}_{k \sim \mathcal{K}} \left[ \mathbb{E}_{\tau_k \sim (\pi, \mathcal{P}_k)} \left[ \sum_{t=0}^{T} \gamma^t r_k(s_{t,k}, a_{t,k}) \right] \right].$$
(12)

**Observation.** Following (Xue et al., 2025), the policy observation $o_t^\pi$ at timestep $t$ consists of a five-step history of proprioception $o_{t-4:t}^P$, a joint controllability indicator $I(t)$ and a whole-body command vector $c_t$. The proprioception observation $o_t^P$ is defined as:

$$o_t^P \triangleq [\omega_t, \ g_t, \ q_t, \ \dot{q}_t, \ a_{t-1}],$$

where $\omega_t \in \mathbb{R}^3$ is base angular velocity, $g_t \in \mathbb{R}^3$ is base gravity direction, $q_t \in \mathbb{R}^{32}$ and $\dot{q}_t \in \mathbb{R}^{32}$ are joint positions and velocities respectively, and $a_{t-1} \in \mathbb{R}^{32}$ is the previous action. The binary indicator $I(t)$ specifies which joints are controllable. The command vector $c_t$ is defined as:

$$c_t \triangleq [\underbrace{v_t^x, v_t^y, \omega_t^z}_{\text{velocity}}, \underbrace{h_t, p_t, \theta_t^y, \theta_t^p, \theta_t^r}_{\text{posture}}, \underbrace{\psi_t, \phi_{t,1}, \phi_{t,2}, \phi_{t,\text{stance}}}_{\text{gait}}],$$
(13)

where $v_t^x$, $v_t^y$ and $\omega_t^z$ are the target base velocity, $h_t$ is the target base height, $p_t$ is the target pelvis angle, $\theta_t^y$, $\theta_t^p$ and $\theta_t^r$ stand for the waist yaw, pitch and roll rotations, and the remaining terms control gait parameters. We further incorporate an upper-body intervention training scheme to control the arm joints for achieving whole-body loco-manipulation tasks. An intervention indicator function $\mathbb{I}(t) = \{0, 1\}$ is introduced to denote whether an external upper-body controller is active. When $\mathbb{I}(t) = 1$, the upper body is driven by an external controller (e.g., teleoperation signals), while the policy still observes these states and adapts lower-body behavior for balance. When $\mathbb{I}(t) = 0$, the upper body is controlled by our whole-body controller. This command space ensures flexible whole-body control for versatile locomotion (Xue et al., 2025). To incorporate the structural information encoded by the robot's graph description, we explore two encoder architectures: Graph Convolutional

*Table 1.* **Average command tracking errors and survival rates aggregated across all robots.** We compare `XHugWBC` with specialist (HugWBC (Xue et al., 2025)) denoting the upper-bound performance, which is trained and evaluated on specific robots separately.

| | Survival | Task Commands | | | Behavior Commands | | | | |
|---|---|---|---|---|---|---|---|---|---|
| | $E_{\text{surv}}(\%)$ | $E_{v_x}(\text{m/s})$ | $E_{v_y}(\text{m/s})$ | $E_\omega(\text{rad/s})$ | $E_{\text{h}}(\text{m})$ | $E_{\text{p}}(rad)$ | $E_{\theta_y}(\text{rad})$ | $E_{\theta_p}(\text{rad})$ | $E_{\theta_r}(\text{rad})$ |
| Specialists (Xue et al. (2025)) | 100 | 0.060 $_{(\pm 0.043)}$ | 0.079 $_{(\pm 0.039)}$ | 0.121 $_{(\pm 0.057)}$ | 0.044 $_{(\pm 0.022)}$ | 0.138 $_{(\pm 0.089)}$ | 0.062 $_{(\pm 0.041)}$ | 0.037 $_{(\pm 0.026)}$ | 0.048 $_{(\pm 0.025)}$ |
| Generalist (XHugWBC) | 100 | 0.084 $_{(\pm 0.042)}$ | 0.116 $_{(\pm 0.022)}$ | 0.160 $_{(\pm 0.021)}$ | 0.044 $_{(\pm 0.023)}$ | 0.097 $_{(\pm 0.039)}$ | 0.043 $_{(\pm 0.028)}$ | 0.052 $_{(\pm 0.027)}$ | 0.049 $_{(\pm 0.008)}$ |

Networks (GCN) (Kipf & Welling, 2017) and Transformers (Vaswani et al., 2017). These encoders model kinematic neighborhoods and produce node features that capture the topology of embodied motion.

**GCN policy encoder.** To model each robot's structure, we employ a Graph Convolutional Network (GCN) (Kipf & Welling, 2017) as a relational encoder. It operates on the node features $\mathbf{X}$ of the robot's connection graph (see Appendix A.3 for details). We stack multiple GCN layers to progressively aggregate information from local kinematic neighborhoods to higher-order relational contexts, producing structure-aware node features.

**Transformer policy encoder.** Transformers provide an alternative means for modeling robot structure via sequence-based attention. The input consists of node embeddings augmented with positional encodings:

$$\mathbf{X}_{\text{pos}} = \mathbf{X} + \mathbf{W}_{\text{pos}} , \tag{14}$$

where $\mathbf{W}_{\text{pos}} \in \mathbb{R}^{N_{\max} \times D}$ are the learned positional embeddings. To exploit kinematic structure, we adopt a topology-aware hybrid-mask strategy: the first layer applies masked attention according to the graph, while subsequent layers use unmasked self-attention for global information exchange. This produces topology-aware attention patterns, enabling integration of both local kinematic structure and global coordination. Additional details are provided in Appendix A.3.

**State estimator.** To mitigate partial observability on real robots, we concurrently train a state estimator to reconstruct privileged information such as base linear velocity and base height. The estimator is optimized via supervised regression, and its outputs serve as the reconstructed privileged information used by the actor detokenizer. This supervised objective is jointly optimized with the RL loss.

**Action prediction.** The node features produced by the encoders are concatenated with a global vector $o_t^g = [w_t, g_t, c_t]$ and the reconstructed privileged information from the learned state estimator. This fused representation is fed into linear layers to generate per-node joint actions. These node-wise actions are then aggregated into a global action vector and mapped back to the robot's physical joints through an embodiment-specific inverse mapping function $\text{inv}(\phi_r) : \mathbb{R}^{N_{\max}} \to \mathbb{R}^{N_r}$.

**Critic structure.** The critic network mirrors the actor but omits the state estimator. Its detokenizer outputs a value estimate for each joint node, and the final value is computed by averaging across node-wise estimates. During training, the critic additionally receives privileged observations that are unavailable on physical hardware, including pelvis and torso linear velocities, torso height, link-collision pairs, and the robot's morphological parameters, to ensure stable and accurate value estimation.

Further architectural details for both the policy and critic networks are provided in Appendix A.3.

## 4. Experiments

We conduct comprehensive evaluations of `XHugWBC` in both simulation and the real-world environments, guided by the following research questions:

**RQ1.** How well does `XHugWBC` generalize to previously unseen embodiments?

**RQ2.** How well can the generalist policy serve as an initialization for fine-tuning on each robot?

**RQ3.** How does the proposed framework compare with cross-embodiment baselines?

**RQ4.** Which policy architecture is more effective for building a generalist humanoid controller?

**RQ5.** How does robot performance differ between simulation and real-world deployment?

### 4.1. Evaluation on Unseen Robots

For RQ1, we evaluate how the generalist policy learned by `XHugWBC` performs on unseen embodiments. Table 1 summarizes results across 12 robots excluded from training, compared with specialist policies trained on each robot. Table 7 provides detailed per-robot results.

The generalist policy demonstrates strong zero-shot generalization across diverse humanoid robots with substantial variations in kinematics, dynamics, and morphology. All robots evaluated achieve a 100% survival rate and maintain consistently high command-tracking accuracy, without exhibiting bias toward any specific system. While specialist policies remain an upper bound for each individual robot,

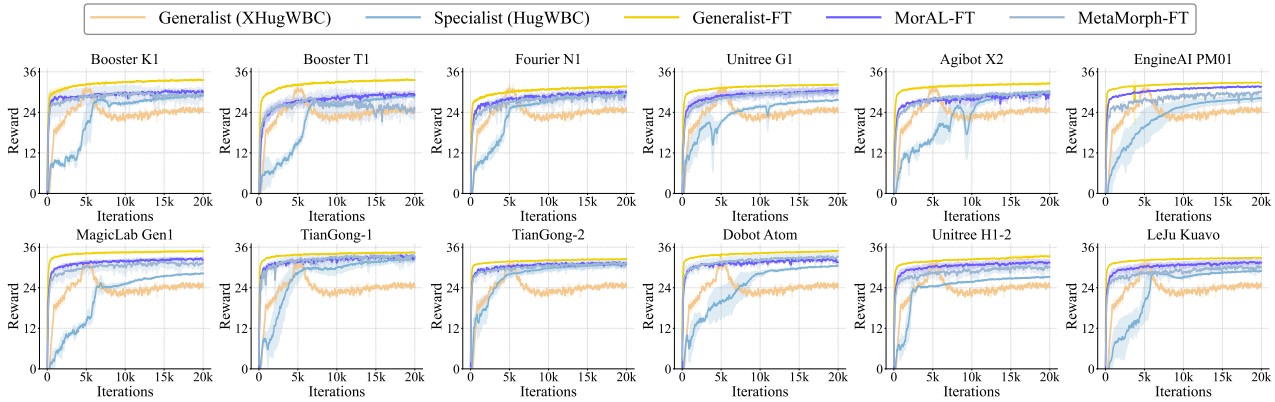

*Figure 3.* **Comparing training curves** of the fine-tuned policies with the generalist policy and specialist policies.

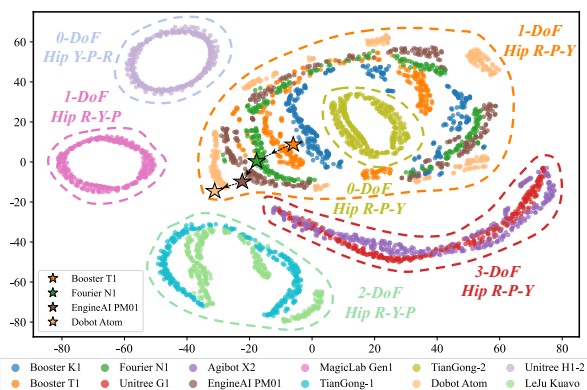

*Figure 4.* **t-SNE visualization of transformer latent representation.** "-DoF" denotes the number of waist joints DoFs. The arrow (←) indicates the direction of increasing robot mass.

XHugWBC achieves comparable performances across embodiments, with expected trade-offs due to lack of system-specific priors. Nevertheless, the generalist provides a powerful pretrained initialization, as discussed in Section 4.2.

**Qualitative Analysis.** To gain insight into how XHugWBC works, we visualize the latent representations produced by the trained transformer before detokenization. Figure 4 shows the t-SNE visualization of the latent vectors. Here, we denote the ordering of the humanoid hip joints, such as **R**oll, **P**itch, and **Y**aw, as **R-P-Y**, with other configurations defined similarly. The results indicate that robot structure shapes the embedded latent distribution: robots with similar hip-joint arrangements cluster closely together.

The number of waist DoFs further modulates the embedding. For example, among robots with the **R-P-Y** hip joint configuration, the robot in yellow (with text **0-DoF**) has no actuated waist joints and appears near the center of the upper-right ring-shaped cluster. Robots with a single waist DoF (orange, with text **1-DoF**) form an outer ring, with lighter robots positioned slightly inward (shown in ⋆). In contrast, robots with three waist DoFs (red and purple, with text **3-DoF**) lie along the outermost boundary.

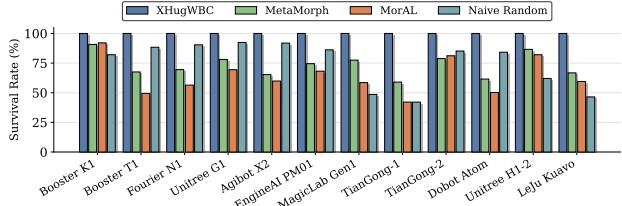

*Figure 5.* **Zero-shot survival rate comparison across multiple baselines.** All policies are trained under the same training protocol. Naive Random is trained on data generated by the naive morphological randomization method, whereas XHugWBC, MetaMorph, and MorAL are trained using randomization in Sec. 3.1.

### 4.2. Fine-tuning the Generalist Policy

To investigate RQ2, we conduct per-robot fine-tuning experiments by reusing the learned weights. Figure 3 compares the training curves of i) generalist policy (XHugWBC), ii) embodiment-specific specialist policies ((Xue et al., 2025)), and iii) the fine-tuned generalist policies (Generalist-FT) across 12 embodiments. At convergence, the generalist policy reaches approximately 85% of the return achieved by the specialist. A temporary drop in performance is observed during training, which arises from the curriculum schedule that progressively increases task difficulty.

On the other hand, fine-tuning the generalist policy yields significantly higher sample efficiency than training from scratch, and converges substantially faster than the other approaches, and fine-tuned generalist rollouts show an approximate 10% improvement in return compared to specialist ones. These results indicate that the learned representations of the generalist are highly adaptable, serving as a strong initialization for efficient specialization while retaining the potential for high-performance control across diverse robots. Table 7 reports the tracking errors for all three policies, showing consistent findings.

### 4.3. Comparing Cross-Embodiment Baselines

For RQ3, Figure 5 also compares XHugWBC with two *cross-embodiment training* baselines, **MetaMorph** (Gupta et al.,

*Table 2.* **Quantitative evaluation of sim-to-real performance gaps.** We zero-shot deploy `XHugWBC` on seven robots and compare average forward tracking errors between simulation and real-world, additionally computing the relative gap to quantify the discrepancy.

| Metrics | Booster T1 | Fourier N1 | Unitree G1 (23 DoF) | Unitree G1 (29 DoF) | Agibot X2 | Dobot Atom | Unitree H1-2 |
|---|---|---|---|---|---|---|---|
| Simulation (m/s) | 0.052 | 0.064 | 0.051 | 0.047 | 0.061 | 0.063 | 0.065 |
| Real-World (m/s) | 0.069 | 0.093 | 0.065 | 0.062 | 0.098 | 0.131 | 0.117 |
| Relative Gap (%) | 32.69 | 45.31 | 27.45 | 31.91 | 60.61 | 107.93 | 80.00 |

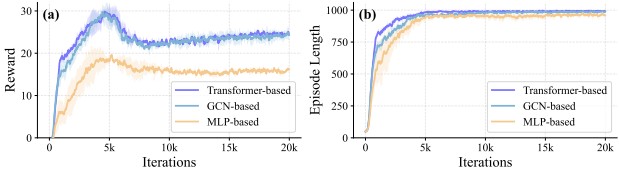

*Figure 6.* **Network architecture ablation.** Training curves comparing MLP-, GNN-, and Transformer-based policies, reporting mean episodic return and mean episode length.

2022) and **MorAL** (Luo et al., 2024) (with the proposed morphology randomization); along with *naive morphology randomization* (Feng et al., 2023; Ai et al., 2025)(**Naive Random** in short) that approximates the rigid body of the robot as a box-shaped primitive, on zero-shot embodiment generalization. Naive random is trained with the same Transformer architecture as `XHugWBC`. MetaMorph's policy relies on explicitly provided morphological information as input, including link inertial properties and joint motion ranges, and it is hard to learn generalizable representations from the high-dimensional, heterogeneous morphological descriptors of humanoid robots. MorAL performs even worse, with a substantial drop in performance observed on most robots, due to the limitations of the simple policy architectures and proprioceptive inputs without informative morphological features. In addition, Naive Random generalizes only to a small subset of embodiments with similar mass distributions and kinematic features, while exhibiting significant performance degradation on the remaining robots.

Figure 3 further reports the fine-tuned results for MetaMorph and MorAL. While fine-tuning both baselines benefit from improved initialization, their convergence is slower than Generalist-FT. In addition, their peak performance is comparable to that of specialist policies, but consistently lower than that achieved by the fine-tuned generalist.

### 4.4. Network Ablation for the Generalist Controller

To answer RQ4, we systematically evaluate different policy network architectures on training convergence and generalization to unseen robots. Figure 6 presents the training curves for MLP-, GNN-, and Transformer-based policies, where **(a)** reports mean episodic return and **(b)** reports mean episode length. Both the GNN and Transformer architectures significantly outperform the MLP baseline. Their per-

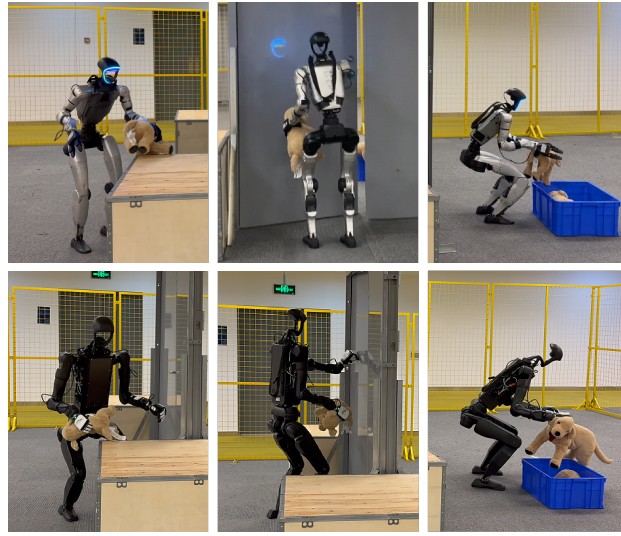

*Figure 7.* **Loco-manipulation sequence: plush toy picking, door opening, and traversal.** The robot first walks toward the box on the right and bends to grasp the plush toy. Next, it opens the door with the other hand, walks through, stops in front of the basket, squats, and places the toy inside, then neatly arranges the toys outside the basket.

formance gain stems from the ability to exploit the robot's embodied kinematic topology and dependency structure encoded in the embodiment adjacency matrix, leading to more sample-efficient learning.

In contrast, even when provided with explicit morphological information (Luo et al., 2024), the MLP struggles to capture inter-joint dependencies. This is because a flattened state discards most of the underlying kinematic structure, making it difficult for the policy to infer meaningful relationships among joints. Based on these results, all subsequent experiments adopt the Transformer architecture.

### 4.5. Sim to Real Performance Gap

We evaluated `XHugWBC` on 7 distinct humanoid robot platforms in real-world environments to test its zero-shot generalization and sim to real performance, as shown in Fig. 1. Hardware specifications of each platform are listed in Table 5. Despite large variations in hardware design, physical properties, and kinematic topology, `XHugWBC` transfers reliably to all platforms and achieves a 100% task success rate.

These results are consistent with simulation.

Due to the lack of external ground-truth systems (e.g., motion capture systems). We introduce a simplified evaluation protocol to quantify the embodiment sim-to-real gap. Specifically, we fix the commanded forward velocity to $v_x = 0.6$ $(m/s)$ over a duration of 10 seconds, and measure the average forward velocity tracking error $E_{\text{error}}$ in both simulation and the real-world settings. To ensure a consistent evaluation protocol across both domains, the average velocity is computed from the traveled distance over time in both simulation and real-world experiments, and the tracking error is defined accordingly. Each experiment is repeated five times, and the reported results are averaged across all trials. This protocol enables a fair and direct comparison between the two domains.

The results (shown in the Tab. 2) reveal that while all platforms exhibit performance degradation in the real world, the magnitude of the relative gap-defined as $\frac{|E_{\text{real}} - E_{\text{sim}}|}{E_{\text{sim}}}$ - varies significantly. Notably, the Booster T1 and Unitree G1 demonstrate superior transferability with relatively small gaps. In contrast, full-size humanoid platforms such as Dobot Atom and Unitree H1-2 show more pronounced discrepancies, which we attribute to hardware factors such as larger gear reduction ratios and increased joint clearances. Interestingly, Fourier N1 and Agibot X2 also exhibit relatively large gaps despite having sizes comparable to Unitree G1, indicating that sim-to-real performance is influenced not only by robot scale but also by platform-specific actuation and mechanical characteristics.

### 4.6. Loco-Manipulation Real World Experiment

Beyond basic locomotion, we further evaluate our method in whole-body control scenarios. Our generalist policy on all robot platforms can robustly and accurately follow full-body commands issued from real-time teleoperation (Cheng et al., 2024b), producing coordinated whole-body behaviors across diverse embodiments, as illustrated in Fig. 1.

We also assess the policy on several long-horizon loco-manipulation tasks using different robots, including Unitree G1 and H1-2 (shown in Fig. 7). In these tasks, each robot is required to approach the box, bend down to pick up the plush toy, open a door, and finally squat down to place the toy into the basket. These tasks demand precise arm control as well as efficient whole-body posture coordination. Despite these challenges, the policy achieves near-perfect success rates across all tasks.

## 5. Conclusion

We present XHugWBC, a scalable cross-humanoid whole-body controller and training framework that enables zero-shot embodiment generalization. Extensive simulation and real-world experiments show that generalist policy transfers robustly across diverse humanoid robots, despite substantial differences in degrees of freedom, dynamics, and kinematic topologies. The learned policy further supports precise and stable control for long-horizon whole-body tasks.

While effective, XHugWBC relies on a unified command interface in which all robots are driven by control signals with shared semantics. This simplifies the learning process but limits applicability to more expressive control settings. For example, motion tracking requires embodiment-specific retargeting, leading to a mismatch between motion representation and robot morphology. Extending cross-embodiment learning to support expressive, morphology-aware control remains an important direction for future work.

## Acknowledgments

This work was supported by the Shanghai Municipal Special Program for Basic Research on General AI Foundation Models (Grant No. 2025SHZDZX025D08), in collaboration with Shanghai Artificial Intelligence Laboratory. The SJTU team is partially supported by National Natural Science Foundation of China (62322603). This work was also supported by the Shanghai Artificial Intelligence Laboratory.

## Impact Statement

This paper presents a policy framework for cross-humanoid whole-body control that enables a single generalist policy to transfer zero-shot across diverse humanoid platforms. Our goal is to improve the scalability of future methods and facilitate rapid transfer and iteration across new embodiments. Notably, the proposed physics-consistent randomization requires embodiment-specific parameter ranges, since different humanoids exhibit distinct physical properties and therefore demand different randomization settings.

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

# A. Implementation Details

## A.1. Proof Skeleton of Proposition 3.5

The parallel-axis theorem establishes a bijective mapping between the rotational inertia w.r.t body frame origin $\bar{\mathbf{I}}$ and $\bar{\mathbf{I}}_C$

$$\bar{\mathbf{I}}_C = \bar{\mathbf{I}} - m\,\mathbf{S}(c)\mathbf{S}(c)^\top . \tag{15}$$

To obtain inertial parameters $\kappa_{\text{link}}$ that corresponds to a physically plausible system (Traversaro et al., 2016; Wensing et al., 2018), we should have constraints:

$$\begin{aligned} m > 0\,, \ D_i > 0,\\ D_1 + D_2 + D_3 > 2D_i \quad \forall i = 1,2,3, \end{aligned} \tag{16}$$

where $m > 0$, $D_1$, $D_2$ and $D_3$ are principal moments of inertia, such that $\bar{\mathbf{I}}_C = \mathbf{R}\mathbf{D}\mathbf{R}^\top$ with $\mathbf{D} = \text{diag}(D_1, D_2, D_3)$ and $\mathbf{R} \in \mathbf{SO}(3)$. Note that the triangle-inequality component of the constraint can be reformulated as:

$$\frac{1}{2}\text{Tr}(\bar{\mathbf{I}}_C) > \lambda_{\max}(\bar{\mathbf{I}}_C)\,, \tag{17}$$

where $\text{Tr}(\cdot)$ is the trace operator, $\lambda_{\max}(\cdot)$ extracts the maximum eigenvalue. Since $\bar{\mathbf{I}}_C$ is a symmetric matrix, it must satisfy $\lambda_{\max}(\bar{\mathbf{I}}_C)\mathbf{I} \succeq \bar{\mathbf{I}}_C$, then Ineq. (17) are equivalent to:

$$\boldsymbol{\Sigma}_C = \frac{1}{2}\text{Tr}(\bar{\mathbf{I}}_C)\mathbf{I} - \bar{\mathbf{I}}_C \succ 0\,. \tag{18}$$

According to the parallel-axis theorem of pseudo-inertia (Wensing et al., 2018):

$$\boldsymbol{\Sigma}_C = \boldsymbol{\Sigma} - m\mathbf{c}\mathbf{c}^\top = \boldsymbol{\Sigma} - \frac{1}{m}\mathbf{h}\mathbf{h}^\top \succ 0\,, \tag{19}$$

according to Schur complement, we can further consolidate the inertial constraints (19) into a linear matrix inequality:

$$\mathbf{J} = \begin{bmatrix} \boldsymbol{\Sigma} & \mathbf{h}^\top \\ \mathbf{h} & m \end{bmatrix} \succ 0\,, \tag{20}$$

where $\mathbf{J} \in \mathbb{R}^{4\times4}$ is pseudo-inertia matrix of rigid body (Rucker et al., 2022), originally defined as the integral form:

$$\mathbf{J} = \int_V \mathbf{q}\mathbf{q}^\top \rho(\mathbf{x})dV\,, \quad \mathbf{q} = [\mathbf{x}^\top, 1]^\top, \tag{21}$$

Therefore, obtaining a physically feasible set of inertial parameters only requires ensuring that the corresponding $\mathbf{J}$ is positive definite. A unique factorization by the Cholesky decomposition claims that a real symmetric matrix $\mathbf{J}$ is positive definite *iff* there exists a unique real nonsingular matrix $\mathbf{L}$ such that

$$\mathbf{J} = \mathbf{L}\mathbf{L}^\top\,,$$

where $\mathbf{L}$ is upper-triangular with a positive diagonal. Now we can physics-consistently randomize the link parameters, by perturbing the upper-triangular entries of $\mathbf{L}$:

$$\mathbf{J}' = \mathbf{L}'\mathbf{L}'^\top\,,\ \mathbf{L}' = \mathbf{L} + \boldsymbol{\epsilon}\,,\ \boldsymbol{\epsilon} \sim \mathcal{D}\,, \tag{22}$$

where $\mathcal{D}$ is the noise distribution. To achieve an interpretable randomization and provide geometric insight into the resulting link, instead of choosing a random distribution $\mathcal{D}$, we model the perturbations as an affine transformation on the rigid bodies' geometry and scaling of the mass density:

$$\begin{aligned} \mathbf{J}' &= \int_V \mathbf{E}\mathbf{q}\mathbf{q}^\top \mathbf{E}^\top \beta^2 \rho(\mathbf{x})dV \\ &= \beta^2 \mathbf{E} \underbrace{\int_V \mathbf{q}\mathbf{q}^\top \rho(\mathbf{x})dV}_{\mathbf{J}} \mathbf{E}^\top \\ &= \mathbf{U}\mathbf{J}\mathbf{U}^\top\,, \end{aligned} \tag{23}$$

where $\mathbf{U} = \beta\mathbf{E}$, $\mathbf{E}$ denotes the linear component of the affine transformation. Equation (23) can be rewritten as:

$$\mathbf{L}'\mathbf{L}'^\top = \mathbf{U}\mathbf{L}\mathbf{L}^\top\mathbf{U}^\top\,. \tag{24}$$

from which it follows that $\mathbf{U} = \mathbf{L}'\mathbf{L}^{-1}$ is a unique upper triangular matrix with positive diagonal entries, since both $\mathbf{L}$ and $\mathbf{L}'$ are upper-triangular. Thus, $\mathbf{U}$ can be parameterized using exponential mapping on the diagonal entries:

$$\mathbf{U} = e^\alpha \begin{bmatrix} e^{d_1} & s_{12} & s_{13} & t_1 \\ 0 & e^{d_2} & s_{23} & t_2 \\ 0 & 0 & e^{d_3} & t_3 \\ 0 & 0 & 0 & 1 \end{bmatrix}\,, \tag{25}$$

we finally obtain a bijective mapping from original inertia to 10-dimensional interpretable vector

$$\theta_{\text{inert}} = [\alpha, d_1, d_2, d_3, s_{12}, s_{23}, s_{13}, t_1, t_2, t_3]^\top \in \mathbb{R}^{10}\,.$$

As a result, the random perturbation $\boldsymbol{\epsilon}$ applied at the Cholesky-factor level can be equivalently expressed as a perturbation in $\theta_{\text{inert}}$. The detailed ranges for $\theta_{\text{inert}}$ can be found in Table 3.

## A.2. Physical Interpretation of Morphological Randomization

Based on the interpretation of $\mathbf{U}$ as a combination of an affine transformation and a density scaling of reference rigid body, each parameter admits a clear physical meaning.

The parameters $d_1, d_2$ and $d_3$ correspond to stretching or compression of the reference rigid body along $x, y$ and $z$ axes, respectively, where $d_i > 0 \Rightarrow e^{d_i} > 1$ implies the stretching, and $d_i < 0 \Rightarrow e^{d_i} < 1$ denotes the compression. Importantly, this exponential formulation guarantees that the body never degenerates to zero thickness.

The shear parameters $s_{12}, s_{13}$ and $s_{23}$ induce shear deformations in the $xy, xz$ and $yz$ planes, respectively, without altering the volume of the object. The translation parameters $t_1, t_2$ and $t_3$ uniformly shift the center of mass of the reference rigid body along the $x, y$ and $z$ directions. Finally, the scalar parameter $\alpha$ controls a global scaling of the mass density $\rho$ by the factor $e^{2\alpha}$.

In this work, we used the Unitree G1 (29 DoF) as the template and added three additional head joints. The corresponding link and joint randomization ranges are listed in Tables. 3 and 4. All noise is sampled from a uniform distribution except for the rotation axis and actuation type. Table 3 defines the randomization ranges for the side lengths of the nominal link bounding box, denoted as $l_x^{\mathrm{ref}}, l_y^{\mathrm{ref}}, l_z^{\mathrm{ref}}$, expressed in the body frame. In the Tab. 4, we denote revolute joint actuation as **R** and fixed joint actuation as **F**. The rotation axes $[a_x, a_y, a_z]$ of the three hip joints are randomly permuted rather than independently sampled. The orientation offsets $[\phi, \theta, \psi]$ are randomly generated with the constraint that their sum across the joint group is zero, ensuring no net orientation bias.

### A.3. Policy Training

To achieve cross-embodiment learning, we employ either GCN or Transformer as representation-learning backbone. Joint-related observations in the policy input $o_t^\pi$, including joint position $q_t$, joint velocities $\dot{q}_t$, actions $a_t$ and $I(t)$, are first mapped to canonical joint representation $\mathbf{q}_{\mathrm{global}}$ via mapping function defined in Equation 10.

The resulting canonical representation is linearly projected, after which each joint node is processed by a node-specific learnable linear layer. This yields a sequence of $N_{\max}$ canonical node embeddings. The remaining policy observations are concatenated into vectors $o_t^g$, which is later used in decoding stage to generate joint-level actions.

Formally, the canonical node embedding $\mathbf{X}$ shared by both encoders. For the Transformer encoder, positional embeddings are added, while the GCN encoder operates directly on $\mathbf{X}$.

$$\mathbf{X} = \left[\phi_\theta^1(\mathbf{q}_{\mathrm{global}}[1]; \mathbf{W}_\theta^1); \cdots ; \phi_\theta^i(\mathbf{q}_{\mathrm{global}}[i]; \mathbf{W}_\theta^i)\right], \quad (26)$$

where the $\phi_\theta(\cdot)$ is learnable embedding function parameters by weights $\mathbf{W}_\theta \in \mathbb{R}^{N \times D}$.

**Encoder.** We then apply a stack of GCN or transformer blocks as the encoder to selectively incorporate information beyond purely local kinematic neighborhood. For the GCN encoder, the input consists of the node embeddings $\mathbf{X}$ (without positional embedding) together with the structural mask $\mathbf{M} = \mathbf{I} + \mathbf{A}$ (with self-loops):

$$\mathbf{Z} = \mathbf{GCN}_{\mathrm{encoder}}(\mathbf{X}, \mathbf{M}), \quad (27)$$

*Table 3.* **Link space parameterization randomization ranges.**

| | Shoulder | Torso | Pelvis | Hip | Knee | Foot |
|---|---|---|---|---|---|---|
| $e^\alpha$ | [0.8, 1.4] | [0.8, 1.5] | [0.8, 1.5] | [0.8, 1.5] | [0.8, 1.5] | [0.8, 1.4] |
| $e^{d_1}$ | [0.8, 1.2] | [0.8, 1.5] | [0.8, 1.4] | [0.8, 1.2] | [0.8, 1.2] | [0.5, 1.5] |
| $e^{d_2}$ | [0.8, 1.2] | [0.8, 1.4] | [0.8, 1.4] | [0.8, 1.2] | [0.8, 1.2] | [0.5, 1.2] |
| $e^{d_3}$ | [0.8, 1.2] | [0.8, 1.2] | [0.8, 1.2] | [0.5, 1.5] | [0.5, 1.5] | [0.8, 1.2] |
| $s_{12}$ | [-0.1, 0.1] | [-0.1, 0.1] | [-0.1, 0.1] | [-0.1, 0.1] | [-0.1, 0.1] | [-0.1, 0.1] |
| $s_{23}$ | [-0.1, 0.1] | [-0.1, 0.1] | [-0.1, 0.1] | [-0.1, 0.1] | [-0.1, 0.1] | [-0.1, 0.1] |
| $s_{13}$ | [-0.1, 0.1] | [-0.1, 0.1] | [-0.1, 0.1] | [-0.1, 0.1] | [-0.1, 0.1] | [-0.1, 0.1] |
| $t_1$ | $[-0.2, 0.2] \times l_x^{\mathrm{ref}}$ | $[-0.2, 0.2] \times l_x^{\mathrm{ref}}$ | $[-0.2, 0.2] \times l_x^{\mathrm{ref}}$ | $[-0.2, 0.2] \times l_x^{\mathrm{ref}}$ | $[-0.2, 0.2] \times l_x^{\mathrm{ref}}$ | $[-0.2, 0.2] \times l_x^{\mathrm{ref}}$ |
| $t_2$ | $[-0.2, 0.2] \times l_y^{\mathrm{ref}}$ | $[-0.2, 0.2] \times l_y^{\mathrm{ref}}$ | $[-0.2, 0.2] \times l_y^{\mathrm{ref}}$ | $[-0.2, 0.2] \times l_y^{\mathrm{ref}}$ | $[-0.2, 0.2] \times l_y^{\mathrm{ref}}$ | $[-0.2, 0.2] \times l_y^{\mathrm{ref}}$ |
| $t_3$ | $[-0.3, 0.3] \times l_z^{\mathrm{ref}}$ | $[-0.3, 0.3] \times l_z^{\mathrm{ref}}$ | $[-0.3, 0.3] \times l_z^{\mathrm{ref}}$ | $[-0.3, 0.3] \times l_z^{\mathrm{ref}}$ | $[-0.3, 0.3] \times l_z^{\mathrm{ref}}$ | $[-0.2, 0.2] \times l_z^{\mathrm{ref}}$ |

*Table 4.* **Joint space parameterization randomization ranges.**

| | Shoulder | Waist | Hip | Knee | Ankle |
|---|---|---|---|---|---|
| $[p_x, p_y, p_z]$ | [0.8, 1.2] | [0.8, 1.2] | [0.8, 1.2] | [0.8, 1.2] | [0.8, 1.2] |
| $[\phi, \theta, \psi]$ | - | - | [-0.3,0.3] (rads) $\sum \delta(\phi, \theta, \psi) = 0$ | - | - |
| $[q_{\min}, q_{\max}, \dot{q}_{\max}]$ | [0.8, 1.0] | [0.8, 1.0] | [0.8, 1.0] | [0.8, 1.3] | [0.8, 1.0] |
| $[a_x, a_y, a_z]$ | - | - | *Permuted* | - | - |
| $\tau_{\max}$ | [0.7, 1.0] | [0.7, 1.0] | [0.7, 1.0] | [0.7, 1.0] | [0.7, 1.0] |
| Actuation | R/F | R/F | R | R | R |

For the Transformer-based encoder, we adopt two types of attention blocks: self-attention blocks (SAB) and masked-attention block (MAB), defined respectively as:

$$\mathbf{SAB} = \mathrm{LayerNorm}(\mathbf{X}_{\mathrm{pos}} + \mathrm{MultiHead}(\mathbf{X}_{\mathrm{pos}}, \mathbf{X}_{\mathrm{pos}})),$$
$$\mathbf{MAB} = \mathrm{LayerNorm}(\mathbf{X}_{\mathrm{pos}} + \mathrm{MultiHead}(\mathbf{X}_{\mathrm{pos}}, \mathbf{X}_{\mathrm{pos}}, \mathbf{M}))$$
$$(28)$$

The hybrid-masking design is similar to prior work (Sferrazza et al., 2024), in which the first encoder layer employs a MAB to respect embodiment-specific structural constraints, while all subsequent layers use SABs to enable global information exchange. Formally, the Transformer encoder is defined as:

$$\mathbf{Z} = \mathbf{Encoder}(\mathbf{X}, \mathbf{M})$$
$$= \mathbf{SAB} \circ \mathbf{SAB} \circ \cdots \circ \mathbf{MAB}. \quad (29)$$

**Action Decoder.** The encoded node features $\mathbf{Z}$ produced by either GCN or Transformer encoder are concatenated with the global context vector $o_t^g$ and the reconstructed privileged information $\mathbf{P}$ obtained from the learned state-estimation module.

$$\mathbf{P} = \mathbf{Estimator}(\mathrm{Flatten}(\mathbf{Z})) \quad (30)$$

where **Estimator** is state estimator trained by supervised learning (Ji et al., 2022; Liu et al., 2025b). Joint-level actions are then decoded independently for each node:

$$\mathbf{a}_{\mathrm{global}} = [\phi_d^1([\mathbf{Z}[1], \mathbf{P}, o_t^g]; \mathbf{W}_d^1); ...\phi_d^i([\mathbf{Z}[i], \mathbf{P}, o_t^g]; \mathbf{W}_d^i)].$$
$$(31)$$

The resulting canonical action vector $\mathbf{a}_{\mathrm{global}}$ is finally mapped back to the robot's physical joints through an embodiment-specific inverse mapping function:

$$\mathrm{inv}(\phi_r) : \mathbb{R}^{N_{\max}} \to \mathbb{R}^{N_r}.$$

*Table 5.* **Robot models used in our experiments**

| Robot type | Evaluation | DoFs | | | Mass (kg) | Height (m) |
| | | Arm | Waist | Leg | | |
| --- | --- | --- | --- | --- | --- | --- |
| Booster K1 | Sim | 4 | 0 | 5 | 18 | 0.95 |
| Booster T1 | Sim / Real | 4 | 1 | 6 | 31 | 1.2 |
| Fourier N1 | Sim / Real | 5 | 1 | 6 | 39 | 1.3 |
| Unitree G1 (23 DoF) | Sim / Real | 5 | 1 | 6 | 33 | 1.4 |
| Unitree G1 (29 DoF) | Sim / Real | 7 | 3 | 6 | 33 | 1.4 |
| Agibot X2 | Sim / Real | 7 | 3 | 6 | 40 | 1.3 |
| EngineAI PM01 | Sim | 5 | 1 | 6 | 40 | 1.4 |
| MagicLab Gen1 | Sim | 7 | 2 | 6 | 66 | 1.7 |
| TianGong-1 | Sim | 4 | 0 | 6 | 42 | 1.7 |
| TianGong-2 | Sim | 4 | 0 | 6 | 61 | 1.7 |
| Dobot Atom | Sim / Real | 7 | 1 | 6 | 60 | 1.7 |
| Unitree H1-2 | Sim / Real | 7 | 1 | 6 | 66 | 1.8 |
| LeJu Kuavo | Sim | 7 | 0 | 6 | 52 | 1.7 |

*Table 6.* **Reward definitions used in `XHugWBC`.**

| Term | Definition | Weight |
| --- | --- | --- |
| **Task Reward** | | |
| Linear Velocity Tracking | $\exp\left(-\left\|v_{xy}^{target} - v_{xy}\right\|^2/0.2\right)$ | 2.5 |
| Angular Velocity Tracking | $\exp\left(-\left\|\omega_z^{target} - \omega_z\right\|^2/0.2\right)$ | 2.0 |
| **Behavior Reward** | | |
| Body Height Tracking | $\left\|h^{target} - h\right\|^2$ | -20 |
| Torso Pitch Tracking | $\left\|p^{target} - p\right\|^2$ | -10 |
| Waist Yaw Tracking | $\left\|\theta_{target}^y - \theta^y\right\|^2$ | -1 |
| Waist Roll Tracking | $\left\|\theta_{target}^r - \theta^r\right\|^2$ | -1 |
| Waist Pitch Tracking | $\left\|\theta_{target}^p - \theta^p\right\|^2$ | -2 |
| Contact-Swing Tracking | $-\sum_i [1 - C(\phi_i)]\left[1 - \exp(\left\|f^{foot,i}\right\|^2/50)\right]$ $-C(\phi_i)\left[1 - \exp\left(\left\|v_{xy}^{foot,i}\right\|^2/5\right)\right]$ | -2 |
| **Regularization Reward** | | |
| R-P Angular Velocity | $\left\|\omega_{xy}\right\|^2$ | -0.5 |
| Vertical Body Movement | $\left\|v_z\right\|^2$ | -0.1 |
| Feet Slip | $1 - \sum_i \exp\left(-\left\|v_{xy}^{foot,i}\right\|^2\right)$ | -0.2 |
| Action Rate | $\left\|a_t - a_{t-1}\right\|^2$ | -0.01 |
| Action Smoothness | $\left\|a_{t-2} - 2a_{t-1} + a_t\right\|^2$ | -0.01 |
| Joint Torque | $\left\|\tau/k_p\right\|^2$ | -5e-6 |
| Joint Acceleration | $\left\|\ddot{q}\right\|^2$ | -2.5e-7 |
| Upper Joint Deviation | $\left\|q_{upper} - q_{upper}^{nominal}\right\|^2$ | -0.5 |
| Head Joint Deviation | $\left\|q_{head} - q_{head}^{nominal}\right\|^2$ | -0.5 |
| Hip Joint Deviation | $\left\|q_{hip,xz} - q_{hip,xz}^{nominal}\right\|^2$ | -1 |
| Zero Actions | $I(t)\left\|a\right\|^2$ | -0.05 |
| Termination | $\mathbb{1}[\text{Early Terminate}]$ | -40 |

## A.4. Reward Details

The total rewards consist of three terms, including task rewards, behavior rewards and regularization rewards. Table 6 lists reward terms for `XHugWBC`. The Contact-Swing Tracking reward is adopted from (Xue et al., 2025).

## B. Extended Experiment

### B.1. Metrics

We evaluate ablation methods in Isaac Gym (Makoviychuk et al., 2021), and the policy performance is measured by two distinct metrics:

**Survival Rate** $E_{surv}$ measures the trajectory survival rate over 1000 environments per episode. Non-termination is defined as the roll and pitch angle of the robot torso do not exceed $80°$.

**Average commands tracking error** $E_{cmd}$ measures the average episodic tracking accuracy for a single command across 1000 environments. The metric is defined as the $L_1$ distance between the actual robot states and the command states. All commands are uniformly sampled within a predefined range.

Figure 8 presents a visualization of simulation evaluation.

Figure 9 shows the embodiment data generated by physics-consistent morphological randomization.

### B.2. Deployment Details

We employ a modular and reusable deployment framework to simplify our sim-to-sim and sim-to-real experiment workflows across diverse embodiments (Lin et al., 2026). During multi-robot experiments, locomotion and manipulation commands are issued from one centralized workstation and synchronized with all robot participants over ZMQ-based communication channels, while each robot executes an identical locomotion policy on its own on-board computing device. In addition, we combine the framework with existing IK solvers (Carpentier et al., 2019; Caron et al., 2025) and teleoperation pipelines (Cheng et al., 2024b; Developers, 2026) to enable whole-body, hardware-agnostic loco-manipulation.

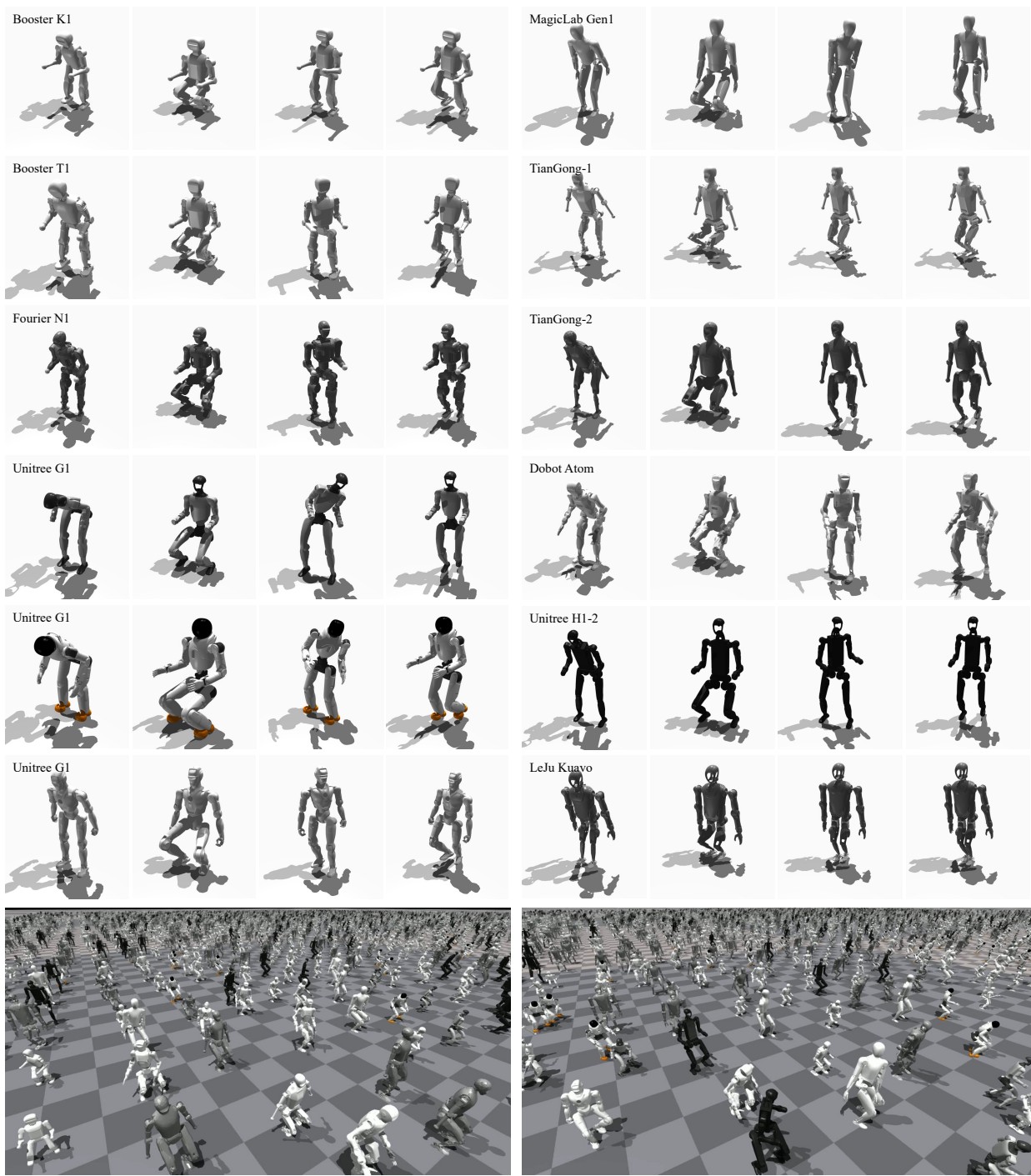

*Figure 8.* **Evaluation of 12 humanoid robots in simulation,** including Booster K1, Booster T1, Fourier N1, Unitree G1, Agibot X2, EngineAI PM01, MagicLab Gen1, TianGong-1, TianGong-2, Dobot Atom, Unitree H1-2, and LeJu Kuavo. The first six rows report per-robot evaluations, while the final rows show parallel evaluations across all robots.

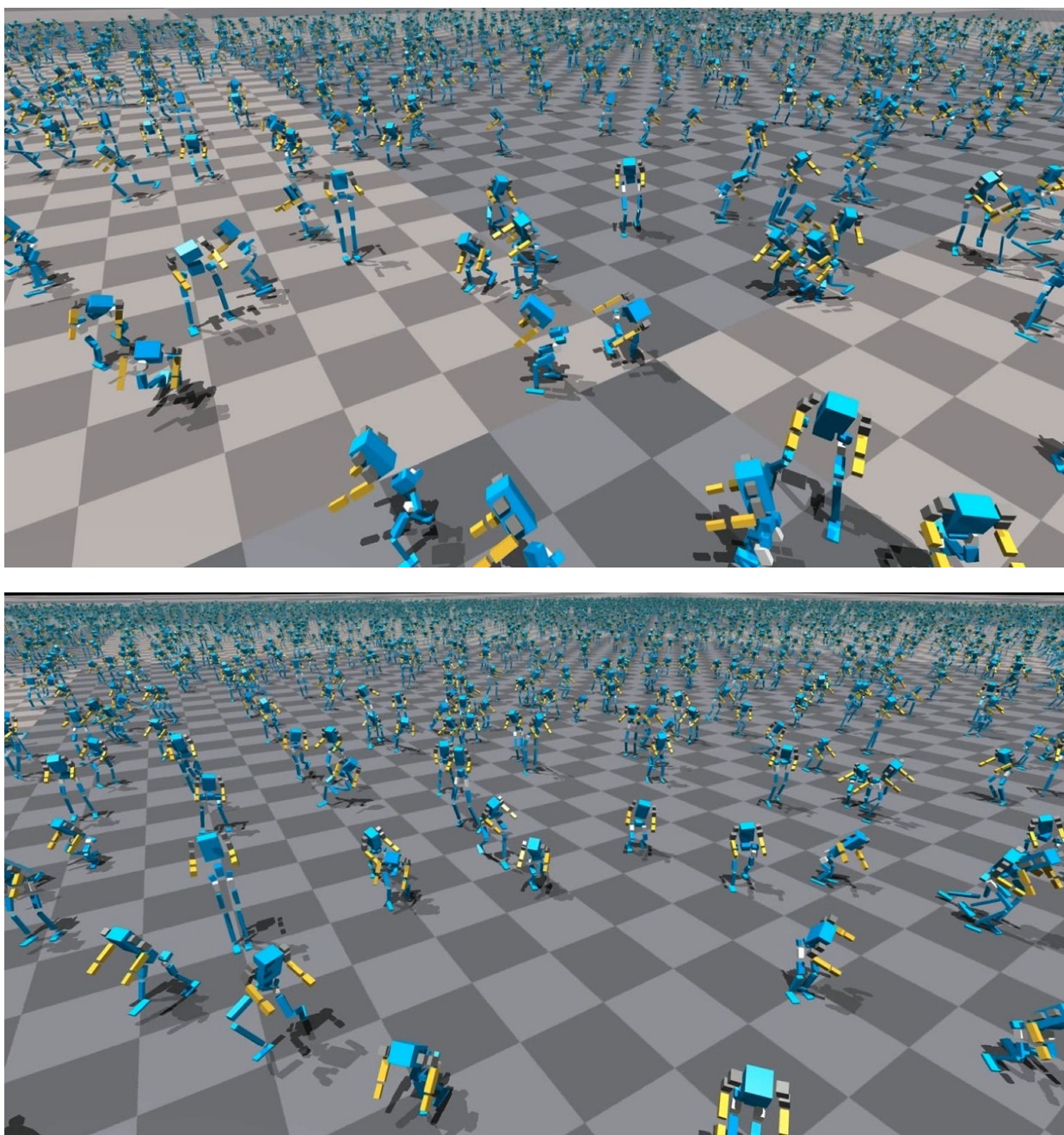

*Figure 9.* **Visualization of samples from procedurally generated embodiment data.**

*Table 7.* **Command tracking errors and survival rates.** Zero-shot performance of XHugWBC, the specialist controller (HugWBC (Xue et al., 2025)) and generalist fine-tuned (Generalist-FT) controller on each of the twelve unseen robots, measured in terms of command tracking errors and survival rates. The symbol "-" indicates that the corresponding robot does not have controllable joints in the specified command space.

| | Survival | Task Commands | | | Behavior Commands | | | | |
|---|---|---|---|---|---|---|---|---|---|
| | $E_{\text{surv}}$ | $E_{v_x}$(m/s) | $E_{v_y}$(m/s) | $E_\omega$(rad/s) | $E_h$(m) | $E_p$(rad) | $E_{\theta_y}$(rad) | $E_{\theta_p}$(rad) | $E_{\theta_r}$(rad) |
| **Generalist (XHugWBC)** | | | | | | | | | |
| Booster K1 | 100 | 0.125 (±0.062) | 0.203 (±0.021) | 0.159 (±0.028) | 0.046 (±0.025) | 0.134 (±0.056) | - | - | - |
| Booster T1 | 100 | 0.094 (±0.034) | 0.142 (±0.027) | 0.281 (±0.013) | 0.075 (±0.043) | 0.090 (±0.021) | 0.039 (±0.025) | - | - |
| Fourier N1 | 100 | 0.090 (±0.040) | 0.119 (±0.031) | 0.177 (±0.011) | 0.015 (±0.006) | 0.112 (±0.034) | 0.044 (±0.026) | - | - |
| Unitree G1 | 100 | 0.047 (±0.069) | 0.079 (±0.023) | 0.145 (±0.021) | 0.067 (±0.036) | 0.094 (±0.036) | 0.041 (±0.034) | 0.042 (±0.030) | 0.038 (±0.006) |
| Agibot X2 | 100 | 0.052 (±0.016) | 0.097 (±0.026) | 0.168 (±0.014) | 0.037 (±0.018) | 0.106 (±0.032) | 0.041 (±0.035) | 0.062 (±0.024) | 0.065 (±0.016) |
| EngineAI PM01 | 100 | 0.075 (±0.029) | 0.107 (±0.022) | 0.150 (±0.013) | 0.039 (±0.026) | 0.142 (±0.043) | 0.044 (±0.027) | - | - |
| MagicLab Gen1 | 100 | 0.106 (±0.031) | 0.086 (±0.022) | 0.151 (±0.030) | 0.031 (±0.016) | 0.076 (±0.052) | 0.054 (±0.038) | - | 0.044 (±0.002) |
| TianGong-1 | 100 | 0.121 (±0.107) | 0.109 (±0.026) | 0.208 (±0.046) | 0.014 (±0.028) | 0.081 (±0.099) | - | - | - |
| TianGong-2 | 100 | 0.094 (±0.033) | 0.155 (±0.016) | 0.146 (±0.011) | 0.038 (±0.015) | 0.119 (±0.023) | - | - | - |
| Dobot Atom | 100 | 0.063 (±0.028) | 0.111 (±0.023) | 0.147 (±0.016) | 0.026 (±0.016) | 0.060 (±0.014) | 0.037 (±0.011) | - | - |
| Unitree H1-2 | 100 | 0.065 (±0.031) | 0.102 (±0.009) | 0.099 (±0.030) | 0.014 (±0.007) | 0.086 (±0.038) | 0.044 (±0.031) | - | - |
| LeJu Kuavo | 100 | 0.075 (±0.025) | 0.082 (±0.026) | 0.098 (±0.024) | 0.126 (±0.040) | 0.072 (±0.027) | - | - | - |
| **Specialist (HugWBC)** | | | | | | | | | |
| Booster K1 | 100 | 0.088 (±0.078) | 0.125 (±0.090) | 0.214 (±0.121) | 0.065 (±0.036) | 0.142 (±0.108) | - | - | - |
| Booster T1 | 100 | 0.029 (±0.011) | 0.091 (±0.017) | 0.191 (±0.012) | 0.057 (±0.002) | 0.101 (±0.055) | 0.059 (±0.035) | - | - |
| Fourier N1 | 100 | **0.057** (±0.050) | 0.076 (±0.046) | 0.140 (±0.101) | 0.052 (±0.039) | 0.173 (±0.098) | 0.057 (±0.032) | - | - |
| Unitree G1 | 100 | 0.054 (±0.049) | 0.061 (±0.036) | 0.110 (±0.076) | 0.054 (±0.035) | 0.121 (±0.070) | 0.050 (±0.030) | 0.045 (±0.026) | **0.036** (±0.023) |
| Agibot X2 | 100 | 0.052 (±0.016) | 0.042 (±0.014) | 0.101 (±0.012) | **0.007** (±0.001) | **0.074** (±0.035) | 0.043 (±0.028) | 0.030 (±0.027) | **0.011** (±0.005) |
| EngineAI PM01 | 100 | **0.030** (±0.014) | 0.074 (±0.009) | 0.109 (±0.023) | 0.061 (±0.001) | 0.096 (±0.054) | 0.072 (±0.044) | - | - |
| MagicLab Gen1 | 100 | 0.090 (±0.054) | 0.080 (±0.048) | 0.130 (±0.087) | **0.029** (±0.021) | 0.179 (±0.121) | 0.059 (±0.030) | 0.097 (±0.046) | 0.097 (±0.046) |
| TianGong-1 | 100 | 0.076 (±0.062) | 0.072 (±0.042) | **0.063** (±0.043) | 0.043 (±0.030) | 0.231 (±0.163) | - | - | - |
| TianGong-2 | 100 | **0.067** (±0.048) | 0.095 (±0.051) | 0.084 (±0.056) | 0.023 (±0.017) | 0.121 (±0.084) | - | - | - |
| Dobot Atom | 100 | 0.061 (±0.044) | 0.065 (±0.038) | 0.145 (±0.086) | 0.034 (±0.022) | 0.126 (±0.083) | 0.118 (±0.107) | - | - |
| Unitree H1-2 | 100 | 0.042 (±0.019) | 0.058 (±0.017) | **0.101** (±0.025) | 0.065 (±0.021) | 0.096 (±0.061) | 0.044 (±0.027) | - | - |
| LeJu Kuavo | 100 | 0.084 (±0.077) | 0.113 (±0.058) | 0.070 (±0.037) | **0.042** (±0.035) | 0.206 (±0.139) | - | - | - |
| **Generalist-FT** | | | | | | | | | |
| Booster K1 | 100 | **0.061** (±0.049) | **0.088** (±0.047) | **0.090** (±0.062) | **0.055** (±0.035) | **0.114** (±0.082) | - | - | - |
| Booster T1 | 100 | **0.027** (±0.009) | **0.088** (±0.016) | **0.181** (±0.002) | **0.033** (±0.021) | **0.084** (±0.011) | **0.029** (±0.015) | - | - |
| Fourier N1 | 100 | 0.061 (±0.049) | **0.068** (±0.039) | **0.135** (±0.089) | **0.039** (±0.029) | **0.096** (±0.045) | **0.031** (±0.019) | - | - |
| Unitree G1 | 100 | **0.039** (±0.021) | **0.058** (±0.031) | **0.091** (±0.055) | **0.051** (±0.036) | **0.102** (±0.051) | **0.026** (±0.014) | **0.033** (±0.017) | 0.037 (±0.022) |
| Agibot X2 | 100 | **0.025** (±0.006) | **0.042** (±0.011) | **0.095** (±0.027) | 0.057 (±0.031) | 0.078 (±0.034) | **0.018** (±0.010) | **0.018** (±0.007) | 0.041 (±0.017) |
| EngineAI PM01 | 100 | 0.043 (±0.005) | **0.051** (±0.006) | **0.091** (±0.018) | **0.016** (±0.013) | **0.082** (±0.038) | 0.035 (±0.012) | - | - |
| MagicLab Gen1 | 100 | **0.064** (±0.051) | **0.068** (±0.035) | **0.103** (±0.064) | 0.033 (±0.023) | **0.121** (±0.063) | **0.024** (±0.014) | - | **0.073** (±0.036) |
| TianGong-1 | 100 | **0.070** (±0.057) | **0.056** (±0.029) | 0.066 (±0.041) | **0.041** (±0.032) | **0.231** (±0.151) | - | - | - |
| TianGong-2 | 100 | 0.079 (±0.049) | **0.091** (±0.051) | **0.075** (±0.087) | **0.018** (±0.026) | **0.080** (±0.055) | - | - | - |
| Dobot Atom | 100 | **0.055** (±0.036) | **0.057** (±0.034) | **0.133** (±0.068) | 0.031 (±0.020) | **0.106** (±0.053) | **0.102** (±0.051) | - | - |
| Unitree H1-2 | 100 | **0.021** (±0.019) | **0.058** (±0.014) | 0.106 (±0.031) | 0.060 (±0.025) | **0.072** (±0.033) | **0.024** (±0.011) | - | - |
| LeJu Kuavo | 100 | **0.061** (±0.055) | **0.099** (±0.051) | **0.058** (±0.034) | 0.048 (±0.040) | **0.158** (±0.113) | - | - | - |

*Table 8.* **Joint index and joint motion direction defined in the global joint space. Each joint axis direction adopts the right-hand rule.**

| Joint Name | Joint Index | Joint Axis |
|---|:---:|:---:|
| Left hip roll | 0 | [1, 0, 0] |
| Left hip pitch | 1 | [0, 1, 0] |
| Left hip yaw | 2 | [0, 0, 1] |
| Left knee pitch | 3 | [0, 1, 0] |
| Left ankle roll | 4 | [1, 0, 0] |
| Left ankle pitch | 5 | [0, 1, 0] |
| Right hip roll | 6 | [1, 0, 0] |
| Right hip pitch | 7 | [0, 1, 0] |
| Right hip yaw | 8 | [0, 0, 1] |
| Right knee pitch | 9 | [0, 1, 0] |
| Right ankle roll | 10 | [1, 0, 0] |
| Right ankle pitch | 11 | [0, 1, 0] |
| Waist pitch | 12 | [0, 1, 0] |
| Waist roll | 13 | [1, 0, 0] |
| Waist yaw | 14 | [0, 0, 1] |
| Head roll | 15 | [1, 0, 0] |
| Head pitch | 16 | [0, 1, 0] |
| Head yaw | 17 | [0, 0, 1] |
| Left shoulder roll | 18 | [1, 0, 0] |
| Left shoulder pitch | 19 | [0, 1, 0] |
| Left shoulder yaw | 20 | [0, 0, 1] |
| Left elbow pitch | 21 | [0, 1, 0] |
| Left wrist roll | 22 | [1, 0, 0] |
| Left wrist pitch | 23 | [0, 1, 0] |
| Left wrist yaw | 24 | [0, 0, 1] |
| Right shoulder roll | 25 | [1, 0, 0] |
| Right shoulder pitch | 26 | [0, 1, 0] |
| Right shoulder yaw | 27 | [0, 0, 1] |
| Right elbow pitch | 28 | [0, 1, 0] |
| Right wrist roll | 29 | [1, 0, 0] |
| Right wrist pitch | 30 | [0, 1, 0] |
| Right wrist yaw | 31 | [0, 0, 1] |

