# OpenReview forum: "Scalable and General Whole-Body Control for Cross-Humanoid Locomotion"
_ICML.cc/2026/Conference — ICML 2026 regular_

### Official Review · Reviewer_eipk · 2026-03-10

**Soundness:** 4
**Presentation:** 3
**Significance:** 3
**Originality:** 3
**Overall Recommendation:** 5
**Confidence:** 2

**Summary:**

The paper proposes a topology-aware transformer policy for cross-embodiment robot locomotion. Robot joints are represented as tokens and a kinematic-tree attention mask restricts interactions based on robot topology. The method trains a shared policy across multiple robot morphologies and uses a learned state estimator to reconstruct privileged variables (e.g., base velocity) from proprioception.

**Compliance With Llm Reviewing Policy:**

Affirmed.

**Final Justification:**

My concerns were addressed well. I will keep my positive score.

**Key Questions For Authors:**

- The state estimator must reconstruct base velocity and height across morphologically randomized robots. Does this mean that it implicitly performs embodiment identification? If so, could providing an explicit embodiment embedding improve estimator accuracy and/or policy performance?
- The topology-aware mask uses binary values to prevent attention between non-adjacent joints. Did the authors explore softer structural biases through continuous attention masks (for example distance-based decay or learned edge weights) that allow limited long-range interactions while still biasing attention toward local kinematic structure?
- GCNs and Transformers could handle cyclic graphs easily. Was collapsing parallel linkages into an acyclic tree just a requirement for the solver? Did you test a version that preserved the parallel linkages/cycles?

Unrelated:
Missing citation on line 65

**Limitations:**

yes

**Strengths And Weaknesses:**

Strengths
- Scales one policy across multiple robot embodiments using a topology-aware architecture.
- Clean integration of transformer tokenization with robot kinematic structure.

Weaknesses
- The reward structure is a fundamental limitation because it applies a unified function with fixed coefficients for torque and acceleration penalties across all training embodiments. While this ensures baseline stability, it creates a performance bottleneck given the physical diversity between robots. A penalty calibrated to stabilize high-inertia systems is disproportionately restrictive for lighter robots, effectively forcing the generalist policy towards a conservative gait.
- A similar performance limitation is introduced through the collapsing of parallel linkages into exclusively parent-child relationships.

---

> ### Author Rebuttal · Authors · 2026-03-30
>
> We thank Reviewer eipk for the insightful feedback and the recognition of our work's contributions to cross-embodiment locomotion.
>
> ## Responses to Weaknesses
>
> ### W1: Reward Structure Limitations
> **Reviewer's Concern:** The unified reward function with fixed coefficients is overly restrictive for all robots, forcing conservative gaits.
>
> **Our Response:**
> Some coefficients are already robot-adaptive, such as torque penalties are normalized by motor gains, and behavioral targets (e.g., base height) scale with robot size. These help account for physical differences, although fully adaptive or embodiment-specific rewards have not yet been explored. The unified reward provides stable training across our robots but may be suboptimal for extremes in mass/inertia. We agree that adaptive rewards or curriculum learning is promising. We clarified this adaptive coefficients setting in the revision.
>
> ### W2: Collapsing Parallel Linkages
>
> **Reviewer's Concern:** Collapsing parallel linkages into parent-child relationships loses performance due to reduced dynamic representation.
>
> **Our Response:**
> We collapse parallel linkages by connecting each joint to its closest common ancestor, motivated by simulation constraints and sim-to-real consistency.
>
> First, simulators do not natively support parallel mechanisms and instead enforce serial approximations, which are often inconsistent with real hardware (e.g., differing joint orderings across platforms). We empirically observe that such simulator-derived graphs may introduce spurious structural biases and degrade transfer performance (e.g., on Fourier N1 and Booster T1).
>
> Second, the physical links in parallel mechanisms are typically short and lightweight relative to upstream links, contributing minimally to overall dynamics. This allows us to approximate these joints as sharing a common parent without losing critical dynamic information. For example, the Agibot X2 ankle parallel joints can be formulated as knee–ankle pitch and knee–ankle roll, instead of knee–ankle pitch–ankle roll.
>
> Although this simplification reduces structural fidelity, we did not observe systematic failures due to missing closed-chain dynamics. In contrast, non-collapsed, simulator-specific graphs tend to harm performance due to their inconsistency with real kinematics. The collapsed, simulator-agnostic representation yields more robust embodiment generalization and sim-to-real transfer. We have added this discussion in the revised appendix.
>
> ## Responses to Questions
>
> ### Q1: State Estimator and Embodiment Identification
>
> **Reviewer's Question:** Does the state estimator implicitly perform embodiment identification? Could explicit embodiment embeddings improve performance?
>
> **Our Response:**
> Correction Regarding Figure 2: The state estimator was misplaced before the attention module. The correct pipeline is: the Transformer/GCN encoder first produces features $\\mathbf{Z}$, then the state estimator reconstructs privileged states from $\\mathbf{Z}$ (this has been corrected in the revision). We hypothesize that the estimator implicitly identifies embodiments from these features $\\mathbf{Z}$, as canonical joint positions, velocities, and actions inherently encode embodiment-specific information. Zero-shot generalization across the 7 robots supports this.
>
> We also tested explicit embodiment inputs (mass, inertia, PD gains) via the Transformer or state estimator. These consistently degraded performance or increased reconstruction loss, suggesting that explicit parameters may distract the model from learning informative structural relationships. We agree with the reviewer that systematic ablation studies are valuable. In response, we have added this discussion in the revised experiment.
>
> ### Q2: Softer Structural Biases via Continuous Attention Masks
>
> **Reviewer's Question:** Did we explore softer structural biases through continuous attention masks (e.g., distance-based decay or learned edge weights)?
>
> **Our Response:**
> We have not yet explored soft masking (e.g., distance-based decay or learned edge weights).
>
> Our approach uses binary masks based on the kinematic tree, providing strong inductive bias, interpretability, and robust zero-shot transfer across embodiments. While this mechanism may be suboptimal for complex parallel linkages or less tree-structured morphologies, it represents a pragmatic choice that works well in practice. We agree that softer structural biases, such as distance-based decay or learned edge weights, are a promising direction, as they can capture graded node relationships, multiple paths, and mechanical couplings. We have added a discussion of this in Sec. 5.
>
> ### Q3: Collapsing Parallel Linkages
>
> This concern is addressed in our response to Weakness W2 above.
>
> ### Q4: Writing and Revision Issues
>
> **Reviewer's Note:** Missing citation on line 65.
>
> **Our Response:**
> The missing citation on line 65 has been added in the revised manuscript.

---

> > ### Author Rebuttal · Reviewer_eipk · 2026-04-03
> >
> > My concerns were addressed by the authors during the rebuttal.

---

### Official Review · Reviewer_1z1o · 2026-03-10

**Soundness:** 4
**Presentation:** 3
**Significance:** 4
**Originality:** 4
**Overall Recommendation:** 4
**Confidence:** 4

**Summary:**

This paper studies the problem of cross-embodiment control for humanoid robots and proposes a framework called XHugWBC that enables a single learned policy to generalize across multiple humanoid platforms. The authors consider a significant challenge in robotics: learning a unified whole-body controller that can operate across robots with different morphologies, kinematics, and dynamics. The proposed approach combines physics-consistent morphological randomization, a unified joint-space representation, and graph/Transformer-based policy architectures to enable cross-humanoid learning. Experiments are conducted on multiple simulated robots and seven real-world humanoids, demonstrating zero-shot generalization and improved fine-tuning efficiency.

**Compliance With Llm Reviewing Policy:**

Affirmed.

**Final Justification:**

My concerns have been addressed. I will keep my positive score.

**Key Questions For Authors:**

The evaluation metrics used in the paper are somewhat limited, as the quantitative analysis mainly focuses on command tracking errors and survival rate. While these metrics provide a useful indication of locomotion stability and basic performance, a more comprehensive evaluation would further strengthen the paper. In particular, a more detailed analysis of **sim-to-real transfer** would be valuable, including discussions of performance gaps between simulation and real-world deployment. In addition, the paper would benefit from experiments on **manipulation-heavy tasks**, where coordinated control between locomotion and manipulation is required. Such tasks would better demonstrate the generality of the proposed approach and its applicability beyond pure locomotion scenarios, providing stronger evidence of the framework’s capability in more complex embodied settings.

**Limitations:**

The above has already explained my question.

**Strengths And Weaknesses:**

# Strength
1. The proposed framework integrates several components including physics-consistent morphological randomization, a unified joint representation, and graph-based policy architectures. The idea of enforcing physical consistency during morphology randomization using pseudo-inertia parameterization and Cholesky decomposition is technically interesting. This improves the realism of randomized embodiments and potentially benefits sim-to-real transfer.

2. The canonical joint-space representation (with a fixed maximum joint dimension) combined with semantic alignment is a pragmatic solution for handling heterogeneous robots with different numbers of joints. The graph-based morphology representation further enables structural information to be incorporated into the policy network, which is well justified for humanoid systems with strong kinematic dependencies.

# Weakness
There are several minor language and formatting issues throughout the paper. For example: “form improved initialization” should be “from improved initialization.” ，“shwon” instead of “shown.”. These errors are minor but should be corrected to improve readability.

---

> ### Author Rebuttal · Authors · 2026-03-30
>
> We thank Reviewer 1z1o for the positive evaluation and helpful suggestions for improving the manuscript.
>
> ## Responses to Weaknesses
> We sincerely apologize for these editorial oversights and thank the reviewer for pointing them out. We have conducted a comprehensive proofreading and revision of the entire manuscript.
>
> ## Responses to Questions
>
> ### Q1: Analysis of Sim-to-Real Transfer
>
> **Reviewer's Request:** A more detailed analysis of sim-to-real transfer, including performance gaps and manipulation-heavy tasks.
>
> **Our Response:**
>
> In the revised manuscript, we have incorporated an additional experiment to quantitatively analyze the sim-to-real performance gap for seven robots.
>
> We note that obtaining precise quantitative measurements on real hardware is inherently challenging due to the lack of external ground-truth systems (e.g., motion capture systems such as MoCap). These constraints make it difficult to directly reproduce simulation-level measurement fidelity in real-world experiments.
>
> To partially address this limitation and provide a more controlled quantitative comparison, we introduce an additional evaluation protocol to quantify the embodied sim-to-real gap. Specifically, we fix the commanded forward velocity to $v\_x=0.6m/s$ over a duration of 10 seconds, and measure the average forward velocity tracking error $E\_\\text{error}$ in both simulation and the real-world setting. To ensure a consistent evaluation protocol across both domains, the average velocity is computed from the traveled distance over time in both simulation and real-world experiments, and the tracking error is defined accordingly. To improve the reliability of the evaluation, each experiment is repeated five times, and we report the mean results. This protocol enables a fair and direct comparison between the two domains.
>
>
> | Robots | Booster T1 | Fourier N1 | Unitree G1 (23DOF) | Unitree G1 (29DOF) | Agibot X2 | Dobot Atom | Unitree H1-2 |
> |:---:|:---:|:---:|:---:|:---:|:---:|:---:|:---:|
> | Simulation (m/s) | 0.052 | 0.064 | 0.051 | 0.047 | 0.061  | 0.063 | 0.065 |
> | Real-World (m/s) | 0.069 | 0.093 | 0.065 | 0.062 | 0.098  | 0.131 | 0.117 |
> | Absolute Gap (%) | 32.69 | 45.31 | 27.45 | 31.91 | 60.61  | 107.93 | 80.00 |
>
> The results (summarized in the table above) reveal that while all platforms exhibit performance degradation in the real world, the magnitude of the absolute gap-defined as $\\frac{E\_\\text{real} - E\_\\text{sim}}{E\_\\text{sim}}$-varies significantly. Notably, the Booster T1 and Unitree G1 demonstrate superior transferability with relatively small gaps. In contrast, full-size humanoid platforms such as Dobot Atom and Unitree H1-2 show more pronounced discrepancies, which we attribute to hardware factors such as larger transmission reducers and increased joint clearances. Interestingly, Fourier N1 and Agibot X2 also exhibit relatively larger gaps despite having sizes comparable to Unitree G1, indicating that sim-to-real performance is influenced not only by robot scale but also by platform-specific actuation and mechanical characteristics.
>
> Regarding manipulation tasks, the current framework primarily optimizes arm joints for high-fidelity position tracking. While this facilitates teleoperated interaction via VR systems, the lack of active force modulation currently limits its effectiveness in manipulation-heavy scenarios. We concur that evaluating tight coordination between locomotion and heavy manipulation is a crucial next step. We have explicitly noted this as a limitation and a focus for future work, where we intend to integrate force-position estimation and feedback mechanisms to enhance the framework’s robustness in complex interactive tasks.

---

> > ### Author Rebuttal · Reviewer_1z1o · 2026-04-06
> >
> > Thank for the response. My concerns have been addressed. I will keep my score.

---

### Official Review · Reviewer_N4KH · 2026-03-12

**Soundness:** 3
**Presentation:** 3
**Significance:** 4
**Originality:** 3
**Overall Recommendation:** 4
**Confidence:** 4

**Summary:**

The authors tackle the problem of cross-embodiment whole-body control (WBC) for humanoid robots. To get a single policy to work across robots with different kinematics, masses, and joint counts, they propose a framework called XHugWBC. The core contributions are threefold: 1) A physics-consistent morphological randomization scheme that parameterizes the pseudo-inertia matrix via Cholesky factorization to ensure rigid-body valid data generation; 2) A mapping of arbitrary humanoid topologies into a unified 32-DoF canonical joint space; and 3) A Transformer-based policy that uses a hybrid attention mechanism to handle the different robot structures. They demonstrate zero-shot transfer to 12 humanoids in simulation and 7 physical humanoids in the real world, and show that this generalist policy serves as a strong initialization for per-robot fine-tuning.

**Compliance With Llm Reviewing Policy:**

Affirmed.

**Final Justification:**

As the authors have adequately addressed my questions, I'd like to maintain my positive score.

**Key Questions For Authors:**

* Q1. In Equation 13, the command vector $c_t$ only contains locomotion, posture, and gait parameters. How exactly are the arm trajectories commanded during the loco-manipulation tasks shown in Figure 7 (e.g., door opening, toy picking)? Are arm commands appended to $c_t$ during these specific tasks, or is the policy just following fixed joint references for the upper body?

* Q2. For the "Naive Random" baseline evaluated in Figure 5, what policy architecture was used? Was it the exact same Transformer setup as XHugWBC, just trained on non-physically-consistent data, or was it a standard MLP?

* Q3. Could you elaborate on the real-world state estimation pipeline? Did you have to train separate supervised state estimators for each of the 7 real-world robots due to differences in sensor suites and noise profiles, or was the state estimator also deployed zero-shot?

* Q4. How is the adjacency matrix constructed for parallel mechanisms (e.g., Agibot)? You mention collapsing parallel linkages to the preceding joint, but doesn't this throw away critical dynamic information about the closed-chain loop? Did you observe failure cases where the collapsed graph was insufficient to represent embodiment-specific behavior?

**Limitations:**

yes

**Strengths And Weaknesses:**

**Strengths:**

* Zero-shot transfer to 7 actual, physical humanoid robots (Unitree, Agibot, Fourier, etc.) is a great empirical achievement. The sim-to-real gap is non-trivial for single humanoids, let alone an ensemble of seven distinct hardware platforms.

* The morphological randomization approach in Section 3.1 is elegant and principled. Naive domain randomization of masses and inertias often results in physically impossible rigid bodies that blow up the physics engine. Reparameterizing the pseudo-inertia matrix using affine transformations guarantees positive definiteness and physically feasible training data. This is a very clean solution.

* The paper is generally well-structured and the architecture ablation (Figure 6) clearly justifies the use of the topology-aware Transformer over MLPs and GCNs.

**Weaknesses:**

* W1. The command space formulation is under-specified for the manipulation tasks. Equation 13 explicitly defines the command vector $c_t$ using only base velocities, torso height, pelvis angles, waist angles, and gait parameters. Yet, Figure 7 shows the robot picking up a toy and opening a door. How are the arm and hand joints being commanded if they aren't in $c_t$? Are they just tracking a hardcoded joint trajectory, or is the policy actually doing whole-body loco-manipulation?

* W2. The method relies on a manually designed 32-DoF canonical mapping (Table 7). While this works for the current crop of humanoids, it requires manual semantic alignment for every new URDF and limits scalability if future robots have multi-DoF spinal joints or complex dexterous hands that exceed the 32-DoF limit.

* W3. The topology-aware Transformer appears to be the strongest architecture in the ablation, although the exact masking scheme is described somewhat inconsistently between the main text and Appendix A.3.

---

> ### Author Rebuttal · Authors · 2026-03-30
>
> We thank Reviewer N4KH for the positive assessment of our work and the insightful questions that have helped improve the clarity of our manuscript.
>
> ## Responses to Weaknesses
>
> ### W1: Command Space Formulation for Manipulation Tasks
> **Reviewer's Concern:** The command vector $c_t$ only contains locomotion/posture parameters, yet Figure 7 shows manipulation?
>
> **Our Response:**
> The arm and hand joints are not directly commanded by $c\_t$.
> For upper body control (arm joints), we employ an external upper-body intervention training following HugWBC. An intervention indicator function $\\mathbb{I}(t) \\in \\{0,1\\}$ denotes whether an external upper-body controller is active. When $\\mathbb{I}(t) = 1$, the upper body is driven by an external controller (e.g., teleoperation signals), while the policy still observes these states and adapts lower-body behavior for balance. When $\\mathbb{I}(t) = 0$, the upper body is controlled by our whole-body controller.
> Hand joints are not modeled in the policy and are executed via predefined or teleoperated trajectories (e.g., VR-based control). We have clarified this intervention training design in Sec. 3.3.
>
> ### W2: Limitation of 32-DoF Canonical Mapping
> **Reviewer's Concern:** The 32-DoF mapping limits scalability for future robots with multi-DoF spines.
>
> **Our Response:**
> This is not a limitation. The 32-DoF canonical space covers current humanoids, and our framework can be readily extended to higher-dimensional settings by increasing $N\_{\\text{max}}$ and adding semantic indices, without modifying the policy architecture. The physics-consistent randomization remains applicable regardless of the number of DoFs.
>
> ### W3: Topology-Aware Transformer Description
> **Reviewer's Concern:** The masking scheme is described inconsistently between the main text and Appendix A.3.
>
> **Our Response:**
> We apologize for the inconsistencies and have corrected them in the revision.
>
> First, Figure 2 misplaced the state estimator before the attention module. It has now been repositioned to operate after the Transformer/GCN encoder.
>
> Second, in Appendix A.3, the definitions of MAB and SAB were inadvertently swapped and have now been corrected. The encoder uses a single MAB in the first layer followed by SABs in subsequent layers.
>
> $$
> \\begin{aligned}
> \\mathbf{MAB} &= \\text{LayerNorm}(\\mathbf{X}\_{\\text{pos}} + \\text{MultiHead}(\\mathbf{X}\_{\\text{pos}}, \\mathbf{X}\_{\\text{pos}}, \\mathbf{M})), \\\\
> \\mathbf{SAB} &= \\text{LayerNorm}(\\mathbf{X}\_{\\text{pos}} + \\text{MultiHead}(\\mathbf{X}\_{\\text{pos}}, \\mathbf{X}\_{\\text{pos}})), \\\\
> \\textbf{Z} & = \\mathbf{Encoder}(\\mathbf{X}, \\mathbf{M}) = \\mathbf{SAB} \\circ \\mathbf{SAB} \\circ \\cdots \\circ \\mathbf{MAB}~.
> \\end{aligned}
> $$
> We have unified the descriptions across Section 3.3 and Appendix A.3, with consistent notation.
>
> ## Responses to Questions
> ### Q1: Upper-Body Control During Loco-Manipulation
> **Reviewer's Question:** How exactly are arm trajectories commanded during loco-manipulation tasks?
>
> **Our Response:**
> As described in W1, we use a hierarchical scheme: arm trajectories are provided via VR teleoperation, while the hands execute predefined grasp primitives.
>
> ### Q2: "Naive Random" Baseline Policy Architecture
> **Reviewer's Question:** What policy architecture was used for the "Naive Random" baseline?
>
> **Our Response:**
> The “Naive Random” baseline uses the same Transformer architecture as XHugWBC. The only difference lies in the training data: it is trained on non-physically-consistent data instead of physics-consistent data. We have clarified this in the revised Sec. 4.2.
>
> ### Q3: Real-World State Estimation Pipeline
> **Reviewer's Question:** Was the state estimator trained separately for each robot, or deployed zero-shot?
>
> **Our Response:**
> The state estimator is deployed zero-shot across all 7 real-world robots, without any robot-specific training.
>
> Correction Regarding Figure 2: The state estimator was misplaced before the attention module. The correct pipeline is: the Transformer/GCN encoder first produces node features $\\mathbf{Z}$, then the state estimator reconstructs privileged states from $\\mathbf{Z}$.
>
> At deployment, proprioception is first mapped to $\\mathbf{q}\_{\\text{global}}$ (Eq. 10). These inputs are then encoded into latent features and processed by the Transformer to generate the node representation $\\mathbf{Z}$. The state estimator reconstructs privileged information from $\\mathbf{Z}$, and the action decoder $\\phi\_{d}$ generates canonical joint-level actions based on $\\mathbf{Z}$, the estimated states, and the global context $o\_t^{g}$, which are then remapped to the specific robot's configuration via the inverse mapping $\\text{inv}(\\phi\_r)$.
>
> ### Q4: Adjacency Matrix for Parallel Mechanisms
>
> **Our Response:**
> This is closely related to Reviewer eipk’s W2. Due to space limitations, we refer the reviewer to that response for a detailed discussion.

---

> > ### Author Rebuttal · Reviewer_N4KH · 2026-04-04
> >
> > Thank you for your detailed response.
> >
> > Just a comment:
> > Q4. I believe the Mujoco simulator should be able to support parallel mechanisms.

---

### Decision · Program_Chairs · 2026-04-30

**Decision:**

Accept (regular)

**Comment:**

This paper presents a single policy that works across humanoid robots with mildly different kinematics, masses, and joint counts. Solid contributions on morphological randomization/alignment and Transformer-based policy that enables zero-shot transfer among humanoids in simulation and real world.

The AC agrees with all reviewers (4,4,5) that this paper reached the bar of ICML.